# FLATQUANT: FLATNESS MATTERS FOR LLM QUANTIZATION

## ABSTRACT

Recently, quantization has been widely used for the compression and acceleration of large language models (LLMs). Due to the outliers in LLMs, it is crucial to flatten weights and activations to minimize quantization error with the equally spaced quantization points. Prior research explores various pre-quantization transformations to suppress outliers, such as per-channel scaling and Hadamard transformation. However, we observe that these transformed weights and activations can still remain steep and outspread. In this paper, we propose FLATQUANT (**F**ast and **L**earnable **A**ffine **T**ransformation), a new post-training quantization approach to enhance flatness of weights and activations. Our approach identifies optimal affine transformations tailored to each linear layer, calibrated in hours via a lightweight objective. To reduce runtime overhead, we apply Kronecker decomposition to the transformation matrices, and fuse all operations in FLATQUANT into a single kernel. Extensive experiments show that FLATQUANT sets up a new state-of-the-art quantization benchmark. For instance, it achieves less than **1**% accuracy drop for W4A4 quantization on the LLaMA-3-70B model, surpassing SpinQuant by **7.5**%. For inference latency, FLATQUANT reduces the slowdown induced by pre-quantization transformation from 0.26x of QuaRot to merely **0.07x**, bringing up to **2.3x** speedup for prefill and **1.7x** speedup for decoding, respectively. Code will be released upon acceptance.

## 1 INTRODUCTION

Recent large language models (LLMs) have achieved remarkable success across a wide range of tasks with an increasing number of parameters (Achiam et al., 2023; Jiang et al., 2023; Yang et al., 2024; Dubey et al., 2024). However, the growth of model size also incurs a significant increase in computation and memory overhead. As a result, reducing the computational and memory demands of LLMs has emerged as a critical research direction, and quantization is one of the most effective solutions (Frantar et al., 2022; Lin et al., 2023; Dettmers et al., 2022; Xiao et al., 2023).

Quantization decreases the memory footprint and accelerates the inference, by reducing the precision of model parameters and activations. Quantization error is a commonly used metric to measure the performance of quantization methods (Nagel et al., 2020; Bai et al., 2020; Li et al., 2021). One key factor that affects the quantization error is the *flatness* of weights and activations. Intuitively, when the distribution of weights and activations is sharp and there exist multiple outspread values, quantizing them to the same quantized value usually incurs a large quantization error (Chmiel et al., 2020; Li et al., 2024). Moreover, as LLMs generate outputs layer by layer, a reduced quantization error also flattens the error landscape propagated across Transformer layers.

Nevertheless, it is non-trivial to get a flat distribution of weights and activations in LLMs. LLMs are known to have extreme outliers over activations (Dettmers et al., 2022; Xiao et al., 2023; Lin et al., 2023) and pivot tokens (Liu et al., 2024a; Sun et al., 2024). To alleviate this problem, various pre-quantization transformations are proposed to mitigate the impact of outliers (Xiao et al., 2023; Ashkboos et al., 2024; Liu et al., 2024b; Ma et al., 2024). However, we revisit these transformations and find them still sub-optimal in promoting flatness. For instance, per-channel scaling (Xiao et al., 2023; Shao et al., 2023) aims to balance the outliers between weights and activations, but it falls short of distributing outliers over the non-outlier channels. The recent Hadamard transformation (Ashkboos et al., 2024; Lin et al., 2024) attempts to solve this problem, while the individual character-

istics of each linear layer are not considered. Moreover, the linear transformation introduced by these methods (Ma et al., 2024; Ashkboos et al., 2024; Liu et al., 2024b) inevitably introduces extra inference overhead that affects the overall speedup of quantization.

In this work, we introduce a new post-training quantization approach named FLATQUANT (**F**ast and **L**earnable **A**ffine **T**ransformation). Our approach is grounded in the principle of achieving a flatter distribution of weights and activations, which is crucial for quantization. FLATQUANT aims to identify the optimal affine transformation for each linear layer, employing a lightweight, block-wise training strategy over the calibration data. To minimize the inference overhead associated with affine transformations, FLATQUANT harnesses the efficiency of Kronecker decomposition, thus reducing both the memory and computational demands. The proposed approach is compatible with various quantization techniques such as learnable clipping, and can be applied to various quantization settings, e.g., weight-only quantization or KV cache quantization. Additionally, by observing that affine transformations in FLATQUANT are memory bound, we further fuse the affine transformations and quantization into a single kernel, thereby minimizing the global memory access and kernel lunch overhead. Lastly, extensive experiments are conducted to assess FLATQUANT across various tasks, including language modeling and question answering, using LLaMA-2/3 models ranging from 7B to 70B parameters. The empirical results demonstrate that our proposed approach surpasses current state-of-the-art methods in terms of both accuracy and inference latency.

The contributions of this work are summarized below:

- We highlight the significance of achieving flatness for LLM quantization, demonstrating that flat distributions of weights and activations facilitate quantization and reduce error propagation across Transformer layers.
- We introduce FLATQUANT, a new post-training quantization method with fast and learnable affine transformations optimized for each linear layer. The approach is empirically demonstrated to enhance the flatness of both weights and activations in LLMs.
- Extensive experiments demonstrate that FLATQUANT sets new state-of-the-art results for quantization. To the best of our knowledge, we are the first to achieve $\leq \mathbf{1}\%$ accuracy drop with simply round-to-nearest W4A4 quantization on the LLaMA-3-70B model.
- We have designed an efficient kernel that fuses affine transformation and quantization, reducing the additional latency caused by transformation from a 0.26x slowdown with QuaRot to only **0.07x**. This enhancement gives up to **2.3x** speedup for prefill and **1.7x** speedup for decoding compared to the FP16 baseline.

## 2 MOTIVATION

### 2.1 PRELIMINARIES ON LLM QUANTIZATION

The inference of LLM typically has two stages: 1) the prefill stage, which creates a key-value cache (KV Cache) layer by layer from the input sequence; and 2) the decoding stage, where the model autoregressively generates tokens based on previous KV cache. Quantization is a common practice to reduce the model size and inference latency. It converts the full-precision weights $\mathbf{W} \in \mathbb{R}^{m \times n}$ or activations $\mathbf{X} \in \mathbb{R}^{k \times n}$ of linear layers (i.e., $\mathbf{Y} = \mathbf{X}\mathbf{W}^\top$), and optionally the KV cache to low-bit representations. For instance, $b$-bit weight quantization can be represented as follows:

$$\hat{\mathbf{W}} = \mathcal{Q}_b(\mathbf{W}) = s \cdot \Pi_{\Omega(b)}(\mathbf{W}/s), \tag{1}$$

where $s$ is the quantization step size, $\Pi(\cdot)$ is the projection function and $\Omega(b) = \{0, 1, ..., 2^b - 1\}$ is the set of $b$-bit integer points. For simplicity of notation, we denote $\mathcal{Q}(\cdot)$ as the general quantization function in the rest of this paper. Quantizing weights enables memory time savings from weight loading from high-bandwidth memory (HBM) into the compute cores, and quantizing activations further reduces the computation, benefiting both the prefill and decoding stages of LLM inference.

As recent works suggest (Xiao et al., 2023; Shao et al., 2023; Xi et al., 2023), LLMs exhibit persistent outliers in activations, posing significant challenges for quantization. Various works have been proposed to suppress these outliers to improve the quantized LLMs. Two most commonly used methods are per-channel scaling (Xiao et al., 2023; Lin et al., 2023; Wei et al., 2023) and Hadamard transformation or its variants (Xi et al., 2023; Ashkboos et al., 2024; Lin et al., 2024).

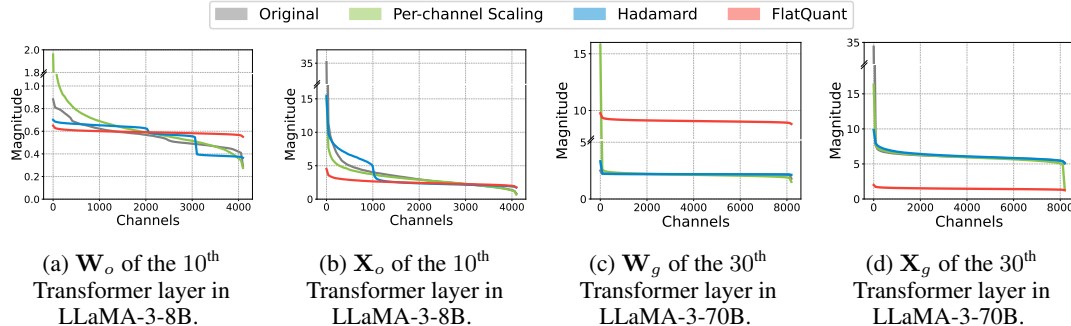

(a) $\mathbf{W}_o$ of the $10^{\text{th}}$ Transformer layer in LLaMA-3-8B.

(b) $\mathbf{X}_o$ of the $10^{\text{th}}$ Transformer layer in LLaMA-3-8B.

(c) $\mathbf{W}_g$ of the $30^{\text{th}}$ Transformer layer in LLaMA-3-70B.

(d) $\mathbf{X}_g$ of the $30^{\text{th}}$ Transformer layer in LLaMA-3-70B.

Figure 1: Distributions of weights and inputs from LLaMA-3-8B and LLaMA-3-70B, sorted by the channel magnitudes (i.e., the Frobenius norm) in descending order. In a Transformer layer, $\mathbf{W}_o$ and $\mathbf{X}_o$ denote the weight matrix and input of the output projection layer in the self-attention layer, respectively. $\mathbf{W}_g$ and $\mathbf{X}_g$ denote the weight and input of the gated linear layer of the feed-forward network, respectively. More visualizations can be found in Appendix D.

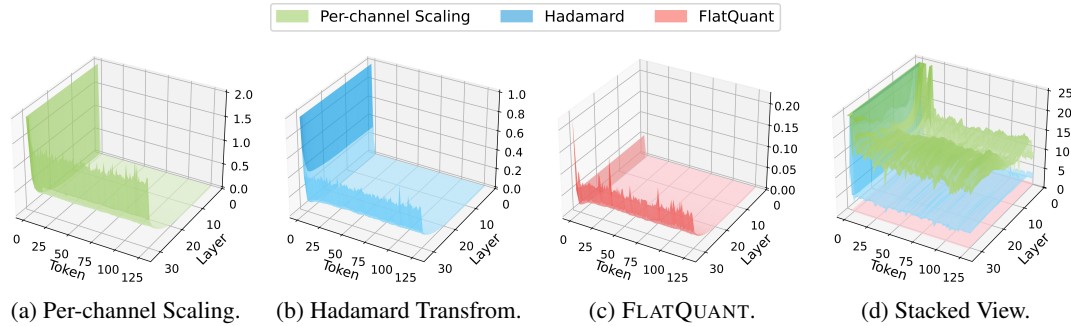

(a) Per-channel Scaling.

(b) Hadamard Transfrom.

(c) FLATQUANT.

(d) Stacked View.

Figure 2: The mean squared error (MSE) of quantization across Transformer layers and input sequence in LLaMA-3-8B. Figure 2a-2c plot the MSE surface of each method, while Figure 2d overlays these surfaces by dividing each MSE with that of FLATQUANT. More details and visualizations can be found in Appendix D.

**Per-channel Scaling.** The input activations $\mathbf{X}$ of LLMs are often rich in outliers. To mitigate their impact on quantization, a popular way is to apply channel-wise scaling over weights and activations (Xiao et al., 2023), i.e., $\mathbf{Y} = (\mathbf{X}\text{diag}(\boldsymbol{c})^{-1}) \cdot (\text{diag}(\boldsymbol{c})\mathbf{W}^\top)$, where $\boldsymbol{c} \in \mathbb{R}^n$ is the channel-wise scaling factor. The scaling vector smooths the activations by jointly considering the magnitudes of input activations and weights, i.e. $\boldsymbol{c}_j = \max(|\mathbf{X}_j|^\alpha)/\max(|\mathbf{W}_j|^{1-\alpha})$. The scaled weights $\text{diag}(\boldsymbol{c})\mathbf{W}^\top$ can be merged to eliminate the runtime computation. Additionally, Wei et al. (2023) introduces channel-wise shifting, i.e., $(\mathbf{X} - \boldsymbol{z})\text{diag}(\boldsymbol{c})^{-1}$, to further mitigate the impact of outliers, and Shao et al. (2023) treats both $\text{diag}(\boldsymbol{c})$ and $\boldsymbol{z}$ as learnable parameters.

**Hadamard Transformation.** Recent works find Hadamard matrices $\mathbf{H} \in \{+1, -1\}^{n \times n}$ are particularly helpful in smoothing out outliers in activations (Xi et al., 2023; Ashkboos et al., 2024; Lin et al., 2024). In contrast to per-channel scaling which only adjusts the diagonal elements in the view of matrix multiplication, Hadamard transformation rotates the channels of both activations and weights, re-distributing the outliers among all channels to effectively eliminate them. Thanks to the orthogonality of Hadamard matrices (i.e., $\mathbf{H}^\top\mathbf{H} = \mathbf{I}$), the following equivalency holds: $\mathbf{Y} = \mathbf{X}\mathbf{W}^\top = (\mathbf{X}\mathbf{H})(\mathbf{H}^\top\mathbf{W}^\top)$. The transformed weight $\mathbf{W}\mathbf{H}$ can be similarly pre-processed offline to reduce additional runtime overhead.

## 2.2 THE FLATNESS FOR QUANTIZATION

We examine existing pre-quantization transformations with a focus on their potential for flatness, a critical factor for effective quantization. Intuitively, by removing outliers, these transformations are expected to produce flat weights and activations that are conducive to quantization. Additionally,

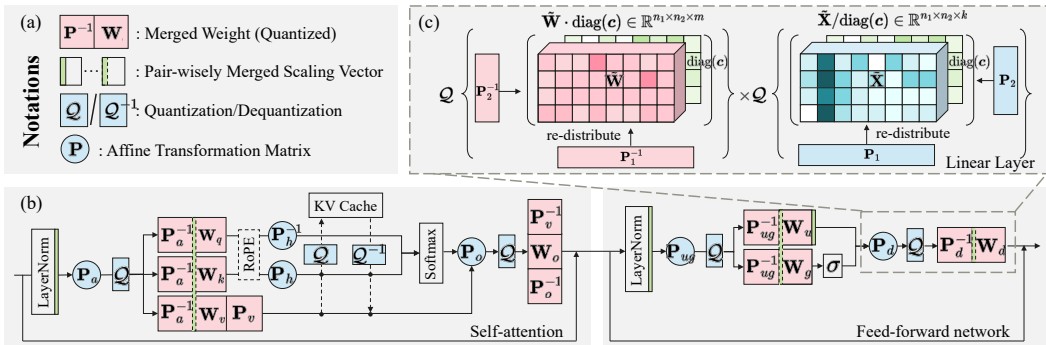

Figure 3: The overall framework of FLATQUANT. (a): necessary notations of FLATQUANT; (b): the integration of FLATQUANT with a conventional LLaMA layer, where merged parameters are grouped in red, online transformation and quantization functions in blue, and merged scaling vectors in green; (c): the exemplary view of FLATQUANT applied for the down-projection layer, where the scaling vector $\text{diag}(c)$ over $\tilde{\mathbf{X}}$ is merged to $\mathbf{W}_u$ in practice.

the quantization error propagated through the network is also expected to be low and flat. However, our empirical results indicate that current methods are limited in achieving the desired flatness. This is in contrast to our proposed FLATQUANT, which learns flatness as discussed in Appendix C.1 and will be introduced in Section 3.

**The Flatness of Weights and Activations.** Flat tensors are intuitively easier to quantize after removing outliers, a.k.a tensors with low kurtosis (Chmiel et al., 2020; Li et al., 2024). Figure 1 displays the distributions of both the original and transformed weights and activations, sorted by the channel magnitudes in descending order. The flat weights and activations with horizontal envelopes are usually preferred by quantization. Compared with the original distributions, pre-quantization transformations can yield flatter activations (e.g., Figure 1b, 1d) but still with their limitations. Per-channel scaling flattens activations at the cost of steeper weight envelops (e.g., Figure 1a, 1c). While Hadamard transformation produces generally better flatness for both activations and weights than per-channel scaling, it still sometimes generates unsatisfactory weights and activations distributions (e.g., Figure 1a, 1b). In contrast, FLATQUANT consistently flattens both weights and activations.

**The Flatness of Quantization Error Landscape.** The quantization error inevitably propagates, and it is insightful to show how pre-quantization transformations mitigate this issue. We plot the two-dimensional landscape of mean squared error (MSE) in Figure 2. First, it is observed that massive quantization errors occur at initial tokens, a.k.a. pivot tokens (Liu et al., 2024a), which contain massive outliers (Sun et al., 2024). Both per-channel scaling and Hadamard transformation are powerless to such errors (i.e., Figure 2a-2b). Instead, FLATQUANT shows much lower error at these pivot tokens from Figure 2c. Second, the quantization error increases layer-wisely, but is less evident along the input sequence. According to Figure 2d, FLATQUANT is the best in controlling the error propagation, followed by Hadamard transformation and lastly the per-channel scaling.

## 3 METHOD

### 3.1 FAST AND LEARNABLE AFFINE TRANSFORMATION

We begin with applying FLATQUANT for a standard linear layer, and will discuss its integration with the Transformer architecture in Section 3.2. The overview of FLATQUANT is presented in Figure 3. A primary objective of FLATQUANT is to find the best affine transformation for each linear layer to quantize. Ideally, given $\mathbf{Y} = \mathbf{X}\mathbf{W}^\top$, one can identify the optimal invertible matrix $\mathbf{P}^* \in \mathbb{R}^{n \times n}$ by

$$\mathbf{P}^* = \arg\min_{\mathbf{P}} \|\mathbf{Y} - \mathcal{Q}(\mathbf{X}\mathbf{P})\mathcal{Q}(\mathbf{P}^{-1}\mathbf{W}^\top)\|_F^2, \qquad (2)$$

as studied in (Ma et al., 2024). The weights $\mathbf{P}^{-1}\mathbf{W}^\top$ can be pre-computed offline akin to (Ashkboos et al., 2024). However, unlike Hadamard matrices that can be reused for all layers, storing

individual $\mathbf{P}$ matrices for different linear layers is computationally expensive. In the forward pass, this approach doubles the computational cost and memory access for matrix multiplication. Additionally, it nearly doubles the model storage requirements. Thus, another key aspect of FLATQUANT is to identify a fast alternative for the pre-quantization transformation.

**Kronecker Decomposition.** We decompose the original $\mathbf{P} \in \mathbb{R}^{n \times n}$ into $\mathbf{P} = \mathbf{P}_1 \otimes \mathbf{P}_2$, where $\mathbf{P}_1 \in \mathbb{R}^{n_1 \times n_1}, \mathbf{P}_2 \in \mathbb{R}^{n_2 \times n_2}$ are invertible matrices in smaller sizes, and $n = n_1 n_2$. Recall the vectorization trick of the Kronecker product, i.e., $\text{vec}(\mathbf{V})(\mathbf{P}_1 \otimes \mathbf{P}_2) = \text{vec}(\mathbf{P}_1^\top \mathbf{V} \mathbf{P}_2)$ for some $\mathbf{V} \in \mathbb{R}^{n_1 \times n_2}$, the matrix multiplication in Equation 2 can be re-written as

$$\mathcal{Q}(\mathbf{X}\mathbf{P})\mathcal{Q}(\mathbf{P}^{-1}\mathbf{W}^\top) = \mathcal{Q}(\mathbf{P}_1^\top \times_1 \tilde{\mathbf{X}} \times_2 \mathbf{P}_2)\, \mathcal{Q}(\mathbf{P}_1^{-1} \times_1 \tilde{\mathbf{W}} \times_2 (\mathbf{P}_2^{-1})^\top)^\top , \tag{3}$$

where $\tilde{\mathbf{X}} \in \mathbb{R}^{k \times n_1 \times n_2}$ and $\tilde{\mathbf{W}} \in \mathbb{R}^{m \times n_1 \times n_2}$ are reshaped from $\mathbf{X}$ and $\mathbf{W}$ accordingly, and $\times_i$ denotes the reduction over the $i$-th axis. Note that both weights and activations are converted back to matrix before multiplication. Such decomposition can save the memory up to $n/2$ times, given that $\frac{n^2}{n_1^2 + n_2^2} \leq \frac{n^2}{2n_1 n_2} = \frac{n}{2}$, with the equality holds when $n_1 = n_2 = \sqrt{n}$. Moreover, the computation saving is $\sqrt{n}/2$ times with the same optimal condition. In practice, we select $n_1^*, n_2^* = \arg\min(n_1 + n_2)$, s.t. $n_1 n_2 = n$ and $n_1 \leq n_2$. For instance, the optimal configuration is $(n_1^*, n_2^*) = (64, 128)$ for $n = 8192$. We find such a strategy gives the best speedup without compromising performance, as will be detailed in Figure 5.

**Per-channel Scaling.** To enhance the ability to balance outliers between the weights and activations, FLATQUANT explicitly introduces a learnable scaling vector $\text{diag}(\boldsymbol{c}) \in \mathbb{R}^n$ prior to the pre-quantization transformation, as illustrated in Figure 3 (c). Following (Xiao et al., 2023), the scaling vector can be merged pair-wisely to the preceding layer normalization or linear layers, thereby incurring no additional inference overhead.

**Learnable Clipping Thresholds.** To further reduce the potential outlier after the above transformation, we combine learnable clipping thresholds $\alpha_w, \alpha_a \in (0, 1)$ on both weight and activation for each linear layer, together with the KV cache. While previous studies (Jacob et al., 2018; Frantar et al., 2022; Ashkboos et al., 2024) demonstrate that grid search is valid to find reasonable clipping thresholds, we observe that learning the clipping thresholds yields better results. These parameters are layer-specific and can be jointly optimized with the linear transformation matrices $\mathbf{P}$ and scaling vector $\text{diag}(\boldsymbol{c})$. A sigmoid function is applied to constrain $\alpha_w$ and $\alpha_a$ within $(0, 1)$.

**The Training Objective.** We are now ready for the training objective of FLATQUANT. We follow post-training quantization and sequentially minimize the mean squared error (MSE) by quantization over a small amount of calibration data (e.g., 128 randomly sampled sentences) for each Transformer block. The training objective for the $l$-th Transformer block is

$$\min_{\Theta}\big\| \mathcal{F}_l(\mathbf{X}) - \hat{\mathcal{F}}_l(\mathbf{X}; \Theta) \big\|_F^2, \tag{4}$$

where $\mathcal{F}_l(\cdot)$ and $\hat{\mathcal{F}}_l(\cdot)$ denote the original and the quantized Transformer block, $\Theta = \{\mathbf{P}, \boldsymbol{c}, \alpha_a, \alpha_w\}$ is abbreviated for all learnable parameters within the block. The transformation matrices within a Transformer block will be explained in Section 3.2. To compute the matrix inversion in Equation 3 efficiently and accurately, we adopt the singular value decomposition together with automatic mixed precision. More details can be found in Appendix B.1. Note that we also experiment with training multiple Transformer blocks together but find similar performance at higher training costs. Finally, we remark that the sequential training with Equation 4 can effectively produce flat weights and activations in Figure 1, and reduce the error propagation along the Transformer blocks in Figure 2.

## 3.2 INTEGRATION WITH THE TRANSFORMER ARCHITECTURE

We illustrate the integration of FLATQUANT with a Transformer block based on an LLaMA-like architecture, as depicted in Figure 3. Following the conventional practices, we employ low-bit matrix multiplications for all linear layers, while keeping layer normalization layers, pre-quantization transformations, RoPE embeddings, and attention scores in FP16.

**Self-Attention.** The self-attention module is equipped with four transformations $\{\mathbf{P}_a, \mathbf{P}_o, \mathbf{P}_h, \mathbf{P}_v\}$. Specifically, $\mathbf{P}_a$ is applied to flatten the input activation for the query, key, and value projections, while $\mathbf{P}_o$ smooths the input activation for the output projection. $\mathbf{P}_h$ and $\mathbf{P}_v$ are used to

transform the key and value cache head by head, respectively. Note that we only decompose $\mathbf{P}_a$ and $\mathbf{P}_o$, but leave $\mathbf{P}_h$ and $\mathbf{P}_v$ in their original shape. This is because per-head quantization already facilitates cheap transformations, given that the head size is significantly smaller than the full hidden size. Moreover, we further fuse $\mathbf{P}_o$ with $\mathbf{P}_v$ to reduce overhead, as inspired by QuaRot (Ashkboos et al., 2024). Our empirical results show this fusion does not result in additional loss of accuracy.

**Feed-forward Network.** The feed-forward network (FFN) employs two transformation matrices, i.e., $\mathbf{P}_{ug}$ and $\mathbf{P}_d$. $\mathbf{P}_{ug}$ is applied to flatten the input of the feed-forward network after layer normalization, while $\mathbf{P}_d$ flattens the input for the down projection layer. Both transformations are decomposed to minimize the inference overhead. Additionally, the per-channel scaling of $\mathbf{P}_d$ is merged into the weight of up projection layer, ensuring no additional computational overhead.

**Layer Normalization.** Recall that QuaRot (Ashkboos et al., 2024) and SpinQuant (Liu et al., 2024b) modify the LayerNorm to RMSNorm and merge orthogonal transformations into preceding layers for efficiency. Nonetheless, the residual connection of the "pre-norm" architecture would constrain all Transformer blocks to share the same transformation after RMSNorm. Instead, FLATQUANT preserves the LayerNorm, and allows the use of fast and learnable affine transformations in Section 3.1 after LayerNorm for different layers, thereby enhancing the expressiveness.

### 3.3 EFFICIENT KERNEL DESIGN

We design an efficient kernel for FLATQUANT that integrates both affine transformations and quantization into a single operation. This design is motivated by two key factors. First, $\mathbf{P}_1^\top \times_1 \tilde{\mathbf{X}} \times_2 \mathbf{P}_2$ exhibits low computational intensity after Kronecker decomposition, making both prefill and decoding predominantly memory-bound. Second, the quantization is also known to be memory-bound.

To address these issues, we fuse $\mathcal{Q}(\mathbf{P}_1^\top \times_1 \tilde{\mathbf{X}} \times_2 \mathbf{P}_2)$ into a single kernel using OpenAI Triton (Tillet et al., 2019). Specifically, we load the entire $\mathbf{P}_1 \in \mathbb{R}^{n_1 \times n_1}$ and $\mathbf{P}_2 \in \mathbb{R}^{n_2 \times n_2}$ into SRAM. Each thread block slices a tiling block $\bar{\mathbf{X}} \in \mathbb{R}^{n_1 \times n_2}$ from $\tilde{\mathbf{X}}$, performs the matrix multiplication $\mathbf{P}_1 \bar{\mathbf{X}} \mathbf{P}_2$, and quantizes the results on the fly. Throughout this process, all intermediate results are stored in SRAM before finally being written back to the global memory. This design thereby eliminates redundant memory accesses of intermediate results and reduces the kernel launch overhead. Further details of the kernel design are provided in the Appendix B.2.

Finally, given the output above, we follow QuaRot (Ashkboos et al., 2024) to adopt the CUTLASS kernel for INT4 matrix multiplication, and FlashInfer (Ye, 2023) for KV cache quantization.

## 4 EXPERIMENTS

### 4.1 SETTINGS

**Evaluation and Baselines.** We evaluate FLATQUANT on the LLaMA-2(7B/13B/70B) (Touvron et al., 2023) models and the LLaMA-3(8B/70B) (Dubey et al., 2024) models. Following previous works (Shao et al., 2023; Ashkboos et al., 2024), we report the perplexity (PPL) of language generation tasks on the WikiText2 (Merity et al., 2016) and C4 (Raffel et al., 2020) datasets. For commonsense reasoning tasks, we use six zero-shot evaluation tasks, including ARC-Challenge, ARC-Easy (Clark et al., 2018), HellaSwag (Zellers et al., 2019), LAMBADA (Paperno et al., 2016), PIQA (Bisk et al., 2020), and WinoGrande (Sakaguchi et al., 2021). Additionally, we evaluate multi-turn conversation ability on LLaMA-3.1-8B-Instruct using MT-Bench, with GPT-4o as the evaluator. For baselines, we compare FLATQUANT against popular INT4 post-training quantization methods, including SmoothQuant (Xiao et al., 2023), OmniQuant (Shao et al., 2023), AffineQuant (Ma et al., 2024), QUIK-4B (Ashkboos et al., 2023), and two recent state-of-the-art methods QuaRot (Ashkboos et al., 2024) and SpinQuant (Liu et al., 2024b).

**Implementation Details.** We implement FLATQUANT based on Huggingface (Wolf, 2019) and PyTorch (Paszke et al., 2019). For optimization, we adopt the AdamW optimizer with an initial learning rate of 5e-3 and employ a cosine annealing learning rate decay schedule. Specifically, the learning rate for clipping thresholds is 5e-2. FLATQUANT is trained for 15 epochs on a calibration set comprising 128 sentences from WikiText-2, each sampled with 2048 tokens. The batch size is

| Method | W Quantizer | WikiText-2 | | | | | C4 | | | | |
|---|---|---|---|---|---|---|---|---|---|---|---|
| | | 2-7B | 2-13B | 2-70B | 3-8B | 3-70B | 2-7B | 2-13B | 2-70B | 3-8B | 3-70B |
| FP16 | - | 5.47 | 4.88 | 3.32 | 6.14 | 2.86 | 7.26 | 6.73 | 5.71 | 9.45 | 7.17 |
| SmoothQuant | RTN | 83.12 | 35.88 | 26.01 | 210.19 | 9.60 | 77.27 | 43.19 | 34.61 | 187.93 | 16.90 |
| OmniQuant | RTN | 14.74 | 12.28 | - | - | - | 21.40 | 16.24 | - | - | - |
| AffineQuant | RTN | 12.69 | 11.45 | - | - | - | 15.76 | 13.97 | - | - | - |
| QuaRot | RTN | 8.56 | 6.10 | 4.14 | 10.60 | 55.44 | 11.86 | 8.67 | 6.42 | 17.19 | 79.48 |
| SpinQuant | RTN | 6.14 | 5.44 | 3.82 | 7.96 | 7.58 | 9.19 | 8.11 | 6.26 | 13.45 | 15.39 |
| FLATQUANT | RTN | **5.79** | **5.12** | **3.55** | **6.98** | **3.78** | **7.79** | **7.09** | **5.91** | **11.13** | **7.86** |
| QUIK-4B | GPTQ | 8.87 | 7.78 | 6.91 | - | - | - | - | - | - | - |
| QuaRot | GPTQ | 6.10 | 5.40 | 3.79 | 8.16 | 6.60 | 8.32 | 7.54 | 6.12 | 13.38 | 12.87 |
| SpinQuant | GPTQ | 5.96 | 5.24 | 3.70 | 7.39 | 6.21 | 8.28 | 7.48 | 6.07 | 12.19 | 12.82 |
| FLATQUANT | GPTQ | **5.78** | **5.11** | **3.54** | **6.90** | **3.77** | **7.86** | **7.11** | **5.92** | **11.21** | **7.93** |

Table 1: WikiText-2 and C4 perplexity of 4-bit weight & acitvation quantized LLaMA models.

set to 4. The default calibration procedure costs approximately 26GB of GPU memory and about 0.9 hours for LLaMA-3-8B on a single GPU. FLATQUANT is robust to initialization, and we employ random affine transformation matrices as the starting point. Further details about implementation and calibration time are provided in Appendix B.1.

**Quantization.** We adopt per-channel and per-token symmetric quantization for weights and activations, respectively. For KV cache quantization, we utilize group-wise asymmetric quantization with a group size of 128. This matches the head dimension of LLaMA, as suggested in previous studies (Zhao et al., 2024; Ashkboos et al., 2024), to effectively leverage the memory-bound characteristics of self-attention. By default, FLATQUANT employs round-to-nearest (RTN) as the weight quantizer. For a fair comparison with QuaRot and SpinQuant, we also report weight quantization using GPTQ, which uses the same calibration data for both closed-form weight updates and training.

## 4.2 MAIN RESULTS

**Results on Language Generation Tasks.** Table 1 presents the PPL results for FLATQUANT with and without the GPTQ weight quantizer on the WikiText-2 and C4 datasets. As can be seen, FLATQUANT with RTN weight quantizer consistently outperforms previous SOTA quantization methods across all major benchmarks. For the LLaMA-2-70B model, FLATQUANT achieves a PPL score just 0.23 higher than the FP16 baseline, underscoring the effectiveness of our approach. For LLaMA-3-8B, FLATQUANT reduces the PPL from 7.39 (SpinQuant) to 6.98, narrowing the gap with the FP16 baseline to 0.84. Notably, FLATQUANT with RTN exhibits performance comparable to those with GPTQ but takes significantly less calibration time. This is particularly helpful in reducing the time consumption to deploy FLATQUANT in practice. These results highlight the efficacy of our proposed learnable transformations in enhancing flatness and mitigating the impact of outliers in both weights and activations, thereby establishing a new SOTA in low-bit LLM quantization.

**Results on Zero-shot QA Tasks.** We extend our evaluation to six zero-shot commonsense QA tasks, as shown in Table 2. For a fair comparison, we reproduce QuaRot [1] and SpinQuant [2] with their official implementations and released checkpoints, evaluating all methods with the same version of lm-eval-harness framework (Gao et al., 2021). As can be seen, FLATQUANT significantly narrows the performance gap between quantized models and the FP16 baseline. Specifically, for larger models such as LLaMA-2-70B, the accuracy loss of FLATQUANT is only 0.43%, which is amazing with such a low bit quantization setting. The recently released LLaMA-3 models have been shown to be more challenging for quantization (Huang et al., 2024). Nonetheless, FLATQUANT continues to perform well, with an accuracy loss of 2.00% for LLaMA-3-8B and 0.94% for LLaMA-3-70B. Notably, while QuaRot with RTN largely lags behind QuaRot with GPTQ by an average accuracy gap over 4%, FLATQUANT with RTN can already obtain comparable results to GPTQ.

**Results on MT-Bench.** We evaluate FLATQUANT on MT-Bench using the LLaMA-3.1-8B-Instruct model in Table 3. While FLATQUANT trails behind the FP16 baseline in coding and

---

[1] https://github.com/spcl/QuaRot
[2] https://github.com/facebookresearch/SpinQuant

| Model | Method | W Quantizer | ARC-C | ARC-E | HellaSwag | LAMBADA | PIQA | Winogrande | Avg |
|-------|--------|-------------|-------|-------|-----------|---------|------|------------|-----|
| **2-7B** | FP16 | - | 46.16 | 74.54 | 75.98 | 73.92 | 79.05 | 69.06 | 69.79 |
| | QuaRot | RTN | 36.60 | 61.41 | 65.07 | 48.06 | 72.20 | 63.06 | 57.73 |
| | SpinQuant | RTN | 39.42 | 65.32 | 71.45 | 66.16 | 75.30 | 63.46 | 63.52 |
| | FLATQUANT | RTN | 43.26 | 72.05 | 73.64 | 72.04 | 77.26 | 69.53 | **67.96** |
| | QuaRot | GPTQ | 42.32 | 68.35 | 72.53 | 65.40 | 76.33 | 65.11 | 65.01 |
| | SpinQuant | GPTQ | 41.72 | 69.28 | 72.90 | 71.28 | 76.17 | 66.06 | 66.23 |
| | FLATQUANT | GPTQ | 43.00 | 71.21 | 73.31 | 72.06 | 77.53 | 67.72 | **67.47** |
| **2-13B** | FP16 | - | 49.15 | 77.44 | 79.39 | 76.73 | 80.47 | 72.14 | 72.55 |
| | QuaRot | RTN | 42.83 | 69.95 | 73.54 | 65.62 | 77.69 | 67.88 | 66.25 |
| | SpinQuant | RTN | 43.69 | 72.43 | 75.52 | 72.42 | 78.40 | 68.90 | 68.56 |
| | FLATQUANT | RTN | 48.04 | 76.64 | 77.59 | 76.60 | 79.38 | 70.24 | **71.42** |
| | QuaRot | GPTQ | 45.48 | 73.27 | 76.03 | 69.01 | 79.05 | 70.64 | 68.91 |
| | SpinQuant | GPTQ | 49.15 | 77.19 | 76.86 | 73.86 | 78.67 | 69.85 | 70.93 |
| | FLATQUANT | GPTQ | 48.38 | 76.94 | 77.88 | 76.40 | 79.65 | 70.56 | **71.64** |
| **2-70B** | FP16 | - | 57.17 | 81.02 | 83.81 | 79.60 | 82.70 | 77.98 | 77.05 |
| | QuaRot | RTN | 52.22 | 76.60 | 79.96 | 74.61 | 81.12 | 76.32 | 73.47 |
| | SpinQuant | RTN | 55.03 | 79.17 | 81.76 | 78.87 | 81.45 | 74.27 | 75.09 |
| | FLATQUANT | RTN | 56.14 | 80.30 | 83.01 | 79.60 | 82.75 | 77.90 | **76.62** |
| | QuaRot | GPTQ | 55.46 | 79.76 | 81.58 | 79.35 | 81.83 | 76.09 | 75.68 |
| | SpinQuant | GPTQ | 55.38 | 79.04 | 82.57 | 78.75 | 82.37 | 78.22 | 76.06 |
| | FLATQUANT | GPTQ | 56.40 | 80.09 | 82.91 | 80.01 | 82.92 | 76.87 | **76.53** |
| **3-8B** | FP16 | - | 53.50 | 77.57 | 79.12 | 75.51 | 80.74 | 72.93 | 73.23 |
| | QuaRot | RTN | 38.65 | 66.54 | 68.82 | 57.20 | 71.82 | 65.04 | 61.34 |
| | SpinQuant | RTN | 45.73 | 71.38 | 74.07 | 67.67 | 76.66 | 66.38 | 66.98 |
| | FLATQUANT | RTN | 50.00 | 75.80 | 76.80 | 72.91 | 79.16 | 72.69 | **71.23** |
| | QuaRot | GPTQ | 45.73 | 70.83 | 72.97 | 62.70 | 75.35 | 67.17 | 65.79 |
| | SpinQuant | GPTQ | 47.27 | 74.20 | 74.55 | 70.29 | 77.37 | 68.51 | 68.70 |
| | FLATQUANT | GPTQ | 50.51 | 75.88 | 76.49 | 73.20 | 79.00 | 72.93 | **71.33** |
| **3-70B** | FP16 | - | 64.25 | 85.94 | 84.93 | 79.37 | 84.44 | 80.74 | 79.95 |
| | QuaRot | RTN | 22.18 | 34.30 | 32.15 | 13.35 | 57.67 | 52.49 | 35.36 |
| | SpinQuant | RTN | 44.03 | 69.07 | 74.57 | 63.34 | 76.99 | 65.98 | 65.66 |
| | FLATQUANT | RTN | 62.12 | 84.97 | 83.95 | 78.73 | 84.28 | 80.03 | **79.01** |
| | QuaRot | GPTQ | 49.49 | 74.37 | 77.22 | 71.69 | 78.89 | 71.03 | 70.45 |
| | SpinQuant | GPTQ | 51.96 | 77.40 | 77.29 | 71.90 | 79.33 | 72.06 | 71.66 |
| | FLATQUANT | GPTQ | 61.95 | 84.47 | 83.87 | 77.99 | 83.95 | 79.24 | **78.58** |

Table 2: Zero-shot QA task results of 4-bit weight & activation quantized LLaMA models.

| Method | Writing | Roleplay | Reasoning | Math | Coding | Extraction | STEM | Humanities | Avg |
|--------|---------|----------|-----------|------|--------|------------|------|------------|-----|
| FP16 | 8.17 | 8.10 | 5.05 | 7.00 | 6.10 | 8.67 | 8.50 | 8.91 | 7.60 |
| QuaRot | 7.20 | 6.90 | 3.90 | 5.30 | 4.05 | 6.70 | 6.05 | 7.80 | 5.99 |
| FLATQUANT | 7.95 | 7.35 | 4.70 | 7.20 | 4.80 | 7.60 | 7.20 | 8.70 | **6.94** |

Table 3: MT-Bench results of 4-bit weight & activation quantized LLaMA-3.1-8B-Instruct model.

STEM tasks, it consistently outperforms QuaRot with GPTQ across all categories, narrowing the gap between the quantized model and the FP16 baseline. Notably, for math problems, FLATQUANT matches the FP16 baseline's score, exceeding QuaRot by 1.9 points. More evaluations are provided in Appendix C.2.

### 4.3 INFERENCE LATENCY

All experiments of inference latency are conducted on the RTX3090 GPU. More details of the overall FLOPs, memory consumption, kernel profiling, and speedup gains are available in Appendix C.5.

**End-to-end Speedup.** Figure 4 shows the prefill and decoding speedup of FLATQUANT across different batch sizes, with 2048 and 256 tokens for prefill and decoding, respectively. With kernel fusion and INT4 tensor core, FLATQUANT can achieve up to 2.30x speedup for prefill and 1.76x speedup for decoding under the batch size of 64. Notably, FLATQUANT is apparently faster than QuaRot (Ashkboos et al., 2024) thanks to the Kronecker decomposition and efficient kernel de-

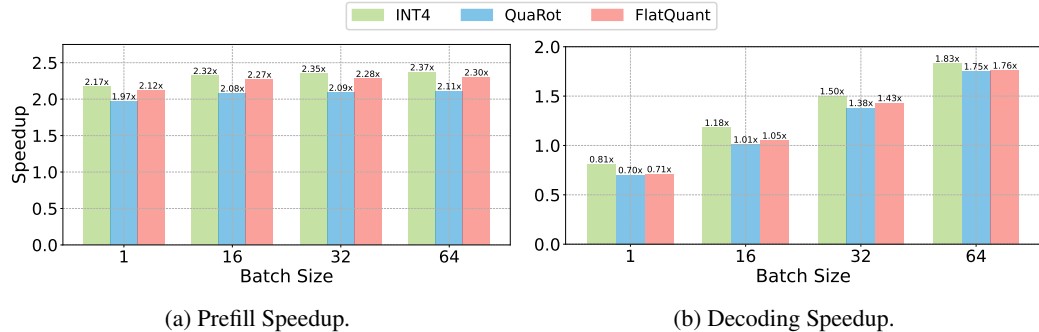

(a) Prefill Speedup.

(b) Decoding Speedup.

Figure 4: Prefill and decoding speedup of LLaMA-2-7B model across different batch sizes. We decode 256 tokens after the prefill on a sequence length of 2048.

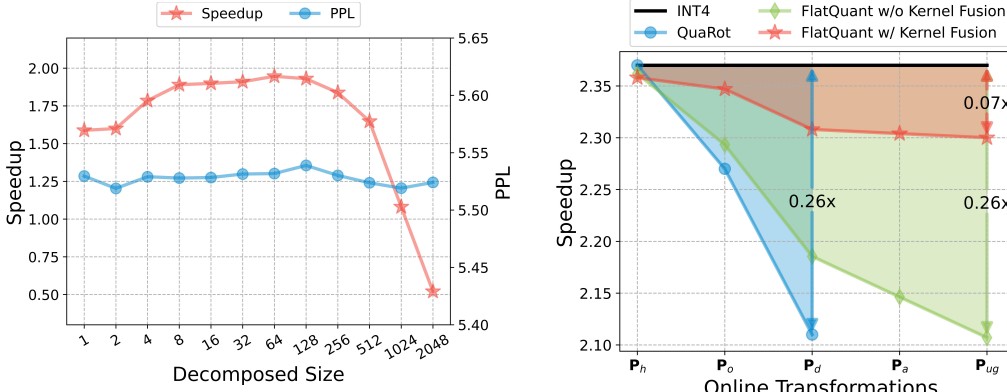

Figure 5: Prefill speedup and WikiText2 PPL results of different decomposed matrix sizes on LLaMA-2-7B model. We decompose the hidden dimension 4096 into $n_1 \times n_2$ and range $n_1$ from 1 to 2048, where $n_1 = 1$ amounts to maintaining a full-size transformation matrix. More details can be found in Appendix C.4.

Figure 6: Prefill speedup of LLaMA-2-7B on a sequence length of 2048 under a batch size of 64 after applying different online transformations. We incorporate different online transformations sequentially to gauge their impact on the final speedup. Each point on the x-axis indicates adding a new online transformation.

sign. Although there is still a minor gap compared to the vanilla INT4 quantization, it significantly enhances accuracy and facilitates the deployment of INT4 LLMs in real-world applications.

**Kronecker Decomposition: Sizes and Perplexities.** In Figure 5, we examine the impact of different decomposed matrix sizes in Equation 3 on model performance and speedup. As shown, the varying sizes of Kronecker decomposition significantly affect speedup, but have limited impact on the perplexity of generated text. The speedup peaks when $\mathbf{P}_1$ and $\mathbf{P}_2$ are of equal size (i.e., $n_1 = n_2 = \sqrt{n} = 64$), as predicted by our theoretical analysis in Section 3.1. When $n_1$ exceeds 64, the speedup quickly decreases due to irregular memory access patterns for activations. These results further demonstrate FLATQUANT's effectiveness in minimizing inference overhead while maintaining quantization accuracy through matrix decomposition.

**Overhead of Each Online Transformation.** We now investigate the impact of the five online transformations (i.e., $\{\mathbf{P}_a, \mathbf{P}_o, \mathbf{P}_h, \mathbf{P}_{ug}, \mathbf{P}_d\}$) in FLATQUANT on the overall speedup, as shown in Figure 6. Even with five per-layer transformations, FLATQUANT results in a minimal 0.07x end-to-end slowdown, significantly outperforming QuaRot's 0.26x with just three Hadamard transformations. Specifically, FLATQUANT's $\mathbf{P}_d$ causes a 0.04x slowdown due to large FFN intermediate sizes, compared with QuaRot's 0.17x. Meanwhile, $\mathbf{P}_o$ results in a 0.01x slowdown, versus QuaRot's 0.1x. The rest transformations (i.e., $\mathbf{P}_a$ and $\mathbf{P}_{ug}$) have an insignificant impact of less than 0.01x.

| LT | PS | LCT | WikiText-2 | C4 | ARC-C | ARC-E | HellaSwag | LAMBADA | PIQA | Winogrande | Avg |
|----|----|-----|-----------|-----|-------|-------|-----------|---------|------|-----------|-----|
| | | | 1266.60 | 936.41 | 25.26 | 28.62 | 27.04 | 1.26 | 51.80 | 51.93 | 30.99 |
| ✓ | | | 8.50 | 13.51 | 44.97 | 71.38 | 73.17 | 67.05 | 76.88 | 67.48 | 66.82 |
| ✓ | ✓ | | 7.95 | 12.74 | 44.20 | 71.89 | 74.21 | 68.72 | 77.15 | 66.30 | 67.08 |
| ✓ | | ✓ | 7.11 | 11.47 | 49.32 | 76.14 | 76.30 | 72.17 | 78.89 | 71.51 | 70.72 |
| ✓ | ✓ | ✓ | 6.98 | 11.13 | 50.00 | 75.80 | 76.80 | 72.91 | 79.16 | 72.69 | 71.23 |

Table 4: Ablation study of FLATQUANT's main components on LLaMA-3-8B.

| LLaMA-3-8B | WikiText-2 PPL | | C4 PPL | |
|------------|------|------|------|------|
| | W4A16 | W3A16 | W4A16 | W3A16 |
| FP16 | 6.14 | | 9.45 | |
| RTN | 8.70 | 2.2E3 | 14.00 | 5.6E3 |
| GPTQ | 7.00 | 13.00 | 11.80 | 45.90 |
| GPTQ-g128 | 6.50 | 8.20 | 10.40 | 13.70 |
| AWQ | 7.10 | 12.80 | 10.10 | 16.80 |
| QuIP | 6.50 | 7.50 | 11.10 | 11.30 |
| FLATQUANT-RTN | 6.54 | 7.78 | 10.17 | 12.64 |
| FLATQUANT-GPTQ | 6.48 | 7.52 | 10.28 | 12.91 |

Table 5: Weight-only quantization results on LLaMA-3-8B model.

| W4 | A4 | KV4 | WikiText-2 PPL | C4 PPL | QA Acc |
|----|----|-----|----------------|--------|--------|
| | | | 6.14 | 9.45 | 73.23 |
| ✓ | | | 6.56 | 10.25 | 72.92 |
| | ✓ | | 6.49 | 10.13 | 72.20 |
| | | ✓ | 6.23 | 9.61 | 73.43 |
| ✓ | ✓ | ✓ | 6.98 | 11.13 | 71.23 |

Table 6: Extending the affine transformations trained under W4A4KV4 to different quantization settings on LLaMA-3-8B model. QA Acc is the average accuray of the six QA tasks in lm-eval-harness.

## 4.4 DISCUSSIONS

**Ablation Study.** We conduct ablation studies for FLATQUANT focusing on its main components: 1) learnable transformation (LT); 2) per-channel scaling (PS); and 3) learnable clipping thresholds (LCT). Starting from RTN as a baseline, we evaluate the impact of each component on perplexity and the average accuracy on zero-shot QA tasks, with LLaMA-3-8B model. As shown in Table 4, enabling LT significantly enhances the accuracy of the quantized model, reducing PPL from 1266.60 to 8.50 on WikiText-2. This shows LT is capable of adaptively flattening the distribution of weight and activation values. Additionally, incorporating PS and LCT further improves PPL by 0.55 and 0.84, respectively, demonstrating the necessity of each component to refine the model performance.

**Other Quantization Schemes.** Although the main results above focus mostly on weight-activation quantization, FLATQUANT can be easily applied to other quantization schemes. The results of weight-only quantization against several state-of-the-art baselines are presented in Table 5. FLATQUANT again obtains leading accuracy compared with leading baselines. For additional results on KV cache quantization and extreme low-bit quantization, please refer to Appendix C.2.

**Train One and Get More.** We demonstrate that the affine transformations learned from weight-activation quantization can be directly applied to other quantization settings, such as weight-only or KV cache quantization, with surprisingly strong performance. The associated results are presented in Table 6. For instance, the results labeled as "W4" are comparable to those in Table 5 that are specifically trained for weight-only quantization. This significantly saves time when applying FLATQUANT to different quantization settings, as only one set of transformation matrices is saved.

Due to space constraints, we provide additional discussions such as the impact of calibration data, and the effect of learnable clipping in Appendix C.3. More visualizations on the flatness of transformed weights and activations, and the quantization error landscapes are in Appendix D.

## 5 CONCLUSIONS

In this study, we revisit the importance of flat weights and activations for effective quantization, and find existing solutions still produce steep outspread values after the pre-quantization transformation. Therefore, we introduce FLATQUANT, a novel post-training quantization method with the purpose of identifying fast and learnable transformations for each linear layer, to promote the flatness of weights and activations. Extensive experiments demonstrate the superiority of FLATQUANT, e.g., with less than **1**% accuracy drop for W4A4 quantization on the LLaMA-3-70B. Our efficient kernel fusion integrates the affine transformation and quantization, reducing the transformation overhead and bringing up to **2.3x** and **1.7x** speedup over FP16 inference at the prefill and decoding stages, respectively. We hope this work advances the practical application of low-bit quantization for LLMs.

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

## A    RELATED WORK

**Quantization for Large Language Models.**    Quantization is a crucial technique for reducing memory footprint and accelerating inference by employing fewer bits for storage and computation, especially for the application of LLMs. Unlike previous models, LLMs are shown to exhibit outliers in activation and massive outliers in pivot tokens (Wei et al., 2022; Dettmers et al., 2022; Liu et al., 2024a; Sun et al., 2024), which can severely degrade quantization accuracy. To eliminate the negative impact of outliers, pre-quantization transformations have been widely adopted in weight-activation quantization (Xiao et al., 2023; Wei et al., 2023; Shao et al., 2023; Ma et al., 2024; Ashkboos et al., 2024; Liu et al., 2024b) as well as in fully quantized training (Xi et al., 2023). Additionally, several weight-only quantization methods (Lin et al., 2023; Chee et al., 2024; Tseng et al., 2024) incorporate pre-quantization transformations. Searched or learnable clipping thresholds on weights or activations (Shao et al., 2023; Lin et al., 2023; Duanmu et al., 2024; Ashkboos et al., 2024; Liu et al., 2024b) are also explored to eliminate outliers.

**Per-channel Scaling Transformation.**    SmoothQuant (Xiao et al., 2023) employs per-channel scaling to shift the challenge of quantization from activations to weights in weight-activation quantization. Building on this, Wei et al. (2023) additionally introduces channel-wise shifting, while OmniQuant (Shao et al., 2023) utilizes a differentiable approach to learn optimal scaling and shifting parameters. However, the scaling-based methods can negatively impact weight quantization and struggle in low-bit settings, such as W4A4 quantization.

**Hadamard and Orthogonal Transformation.**    Recent research (Xi et al., 2023; Tseng et al., 2024; Ashkboos et al., 2024) has shown that the Hadamard transformation is effective in eliminating outliers and lowering quantization error by redistributing outliers across all channels through matrix multiplication. QuaRot (Ashkboos et al., 2024) is the first to apply Hadamard transformation in the LLM W4A4 PTQ setting, while SpinQuant (Liu et al., 2024b) exploits learnable orthogonal matrices with model-level loss to further alleviate outliers.

**Affine Transformation.**    Considering that per-channel scaling corresponds to the diagonal elements of the affine transformation matrix, AffineQuant (Ma et al., 2024) proposes learning the equivalent affine transformation. However, their approach focuses on learning full-size diagonally dominant matrices and employs a gradual mask optimization method, which may hinder the full potential of affine transformation in reducing quantization loss. Moreover, due to the formidable overhead associated with full-sized matrix multiplication, AffineQuant can only apply affine transformation to a small fraction of linear layers. In contrast, we employ fast and learnable affine transformations without these limitations, leading to substantial accuracy improvements and practical speedup.

**Pre-quantization Transformations in Other Quantization Tasks.**    Inspired by SmoothQuant, AWQ (Lin et al., 2023) introduces activation-aware per-channel scaling to reduce quantization errors in weight-only quantization. QUIP (Chee et al., 2024) and its extension, QUIP# (Tseng et al., 2024), leverage random rotation matrices or Hadamard transformations to enhance incoherence in weight-only quantization. In fully quantized training task, Xi et al. (2023) propose to utilize a block-diagonal transformation consisting of Hadamard matrices to reduce the quantization error.

## B    IMPLEMENTATION DETAILS

### B.1    MATRIX INVERSION AND TRAINING COST

A critical aspect to implement FLATQUANT is the computation of the inverse affine transformation matrix $\mathbf{P}^{-1}$. As discussed below, we use singular value decomposition (SVD) and automatic mixed precision to train FLATQUANT, enjoying both training stability and efficiency.

**Direct Inversion and FP32 Training.**    One straightforward approach is to use the inverse function provided by PyTorch. However, we find that the precision of this inverse function at FP16 is insufficient. Specifically, $\mathbf{P}\mathbf{P}^{-1}$ does not closely approximate $\mathbf{I}$. The off-diagonal elements are on the order of $1 \times 10^{-3}$, which negatively impacts FLATQUANT's performance during the early stages

of training. Therefore, a simple solution is to conduct training in FP32 without Automatic Mixed Precision (AMP) to maintain precision. However, this inevitably increases training time and more GPU memory consumption.

**SVD and AMP Training.** To further reduce resource requirements during calibration, we propose to employ singular value decomposition for the affine transformation. For any real matrix $\mathbf{P}$, we can decompose it as $\mathbf{P} = \mathbf{U}\boldsymbol{\Sigma}\mathbf{V}^\top$, where $\mathbf{U}$ and $\mathbf{V}$ are orthogonal matrices, and $\boldsymbol{\Sigma}$ is a diagonal matrix. This formulation allows us to easily compute $\mathbf{P}^{-1} = \mathbf{V}\boldsymbol{\Sigma}^{-1}\mathbf{U}^\top$, offering a more computationally efficient method for obtaining the inverse. Notably, this approach reduces the off-diagonal elements of $\mathbf{P}\mathbf{P}^{-1}$ to the order of $1 \times 10^{-6}$ at FP16 precision, enabling us to utilize AMP during calibration. With AMP, we can achieve a 50% reduction in training time and memory usage while maintaining nearly lossless accuracy in most cases. For the orthogonal matrices $\mathbf{U}$ and $\mathbf{V}$, we employ the Cayley parameterization provided by PyTorch [3].

**Comparison of the Two Training Recipes.** We compare the two training recipes in Table 7. As shown, FP32 training requires more than twice the time of AMP training and necessitates 1.28x more GPU memory under the same setting, while the performance remains relatively close. Thus, our default choice is the SVD approach combined with AMP training. However, we observe that in certain models or extremely low-bit scenarios, numerical errors may occur within the AMP framework. In such cases, full-precision training becomes necessary.

| Training Recipe | | WikiText-2 PPL | C4 PPL | QA Acc | Memory | Time |
|---|---|---|---|---|---|---|
| FP32 | Inverse | 6.95 | 11.04 | 71.35 | 35384MiB | 2.2 hours |
| | SVD | 9.96 | 11.07 | 71.24 | 35360MiB | 2.2 hours |
| AMP | Inverse | 7.00 | 11.17 | 70.57 | 27624MiB | 0.9 hours |
| | SVD | 6.98 | 11.13 | 71.23 | 27554MiB | 0.9 hours |

Table 7: Comparison of different training recipes for FLATQUANT on the LLaMA-3-8B.

**Calibration Time.** We further present the calibration time required by FLATQUANT for the LLaMA family in Table 8. Compared to SpinQuant (Liu et al., 2024b) and QAT methods, FLATQUANT requires significantly fewer computational resources and less training time, while delivering superior performance. For weight-only quantization, only transformations related to the linear weights are introduced, resulting in a shorter calibration time compared to weight-activation quantization. Moreover, as discussed in Section 4.2, FLATQUANT does not need to be combined with GPTQ to achieve optimal performance, further reducing the calibration overhead.

| LLaMA | 2-7B | 2-13B | 2-70B | 3-8B | 3-70B |
|---|---|---|---|---|---|
| weight-activation | 1.15 hours | 1.55 hours | 6.15 hours | 0.90 hours | 5.94 hours |
| weight-only | 0.67 hours | 1.01 hours | 5.00 hours | 0.70 hours | 4.89 hours |

Table 8: Calibration time for LLaMA models. The reported times correspond to training on 128 segments of 2048 tokens over 15 epochs with a batch size of 4, using a single GPU.

## B.2 MORE DISCUSSIONS ON KERNEL FUSION

To avoid redundant memory access and improve computational efficiency, we attempt to fuse $\mathcal{Q}(\mathbf{P}_1^\top \times_1 \bar{\mathbf{X}} \times_2 \mathbf{P}_2)$ into a single kernel, followed by the INT4 CUTLASS kernel to multiply the 4-bit quantized weights and activations. In most cases, the shared memory per thread block is sufficient to hold the source matrices $\mathbf{P}_1$, $\mathbf{P}_2$, $\bar{\mathbf{X}}$, and their intermediate results $\bar{\mathbf{X}}'$, as visualized in Figure 7a. Nonetheless, there are corner cases when the shared memory is insufficient to hold all

---

[3]https://pytorch.org/docs/stable/generated/torch.nn.utils.parametrizations.orthogonal.html

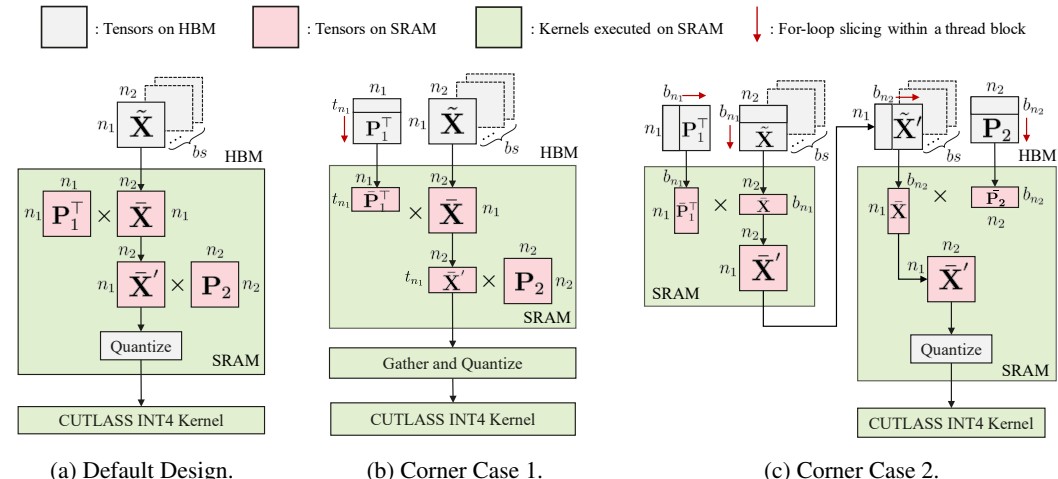

Figure 7: The visualization of the kernel fusion in FLATQUANT based on the computation within a thread block. The design holds mainly for (a), where both transformations and quantization are fused together. For completeness, we also revise the design for corner cases in (b) and (c), when the SRAM is not large enough to hold the intermediate results.

necessary tensors (e.g., $n > 28762$ with $n_1, n_2 > 128$ on the NVIDIA RTX 3090). We thus revise our design for the two cases, as shown in Figure 7b and Figure 7c, respectively. To distinguish these scenarios more clearly, we have the following equations:

$$
\begin{aligned}
\textbf{Default Design:} \quad & (n_1 * n_1 + 2 * n_1 * n_2) * 2 < m \\
& (n_2 * n_2 + 2 * n_1 * n_2) * 2 < m
\end{aligned}
\tag{5}
$$

$$
\begin{aligned}
\textbf{Corner Case 1:} \quad & (t_{n_1} * n_1 + n_1 * n_2 + t_{n_1} * n_2) * 2 < m \\
& (n_2 * n_2 + 2 * t_{n_1} * n_2) * 2 < m
\end{aligned}
\tag{6}
$$

$$
\begin{aligned}
\textbf{Corner Case 2:} \quad & (n_1 * b_{n_1} + b_{n_1} * n_2 + n_1 * n_2) * 2 < m \\
& (n_1 * b_{n_2} + b_{n_2} * n_2 + n_1 * n_2) * 2 < m
\end{aligned}
\tag{7}
$$

where $m$ is the shared memory size per thread block, $t_{n_1}$ is the tiling size of non-reduction dimension of $\mathbf{P}_1$, $b_{n_1}$ is the tiling size of reduction dimension of $\mathbf{P}_1$, $b_{n_2}$ is the tiling size of reduction dimension of $\mathbf{P}_2$ and 2 refers to two bytes to hold tensors in float16. Below we review the designs for the two corner cases respectively.

**Corner Case 1.** When both $n$ and $n_1$ are excessively large, it is suggested to prevent from loading the entire $\mathbf{P}_1$ and $\bar{\mathbf{X}}$ into SRAM. We manage this by tiling the non-reduction dimension of $\mathbf{P}_1$ into $t_{n_1}$ slices. This strategy enables us to integrate $\bar{\mathbf{P}}_1 \bar{\mathbf{X}} \mathbf{P}_2$ into one kernel, with $\bar{\mathbf{P}}_1$ representing a slice of $\mathbf{P}_1$ on the non-reduction dimension. Subsequently, we invoke a separate fused kernel for quantization, computing the quantization scale and scaling the input.

**Corner Case 2.** When both $n$ and $n_2$ are extremely large, $\mathbf{P}_1$, $\bar{\mathbf{X}}$ and $\mathbf{P}_2$ cannot be loaded into SRAM together. To handle this, we first compute $\bar{\mathbf{X}}' = \bar{\mathbf{P}}_1^\top \bar{\mathbf{X}}$, where each thread block slicing the non-reduction dimension of $\mathbf{P}_1$ and $\bar{\mathbf{X}}$ with the tiling shape $b_{n_1}$. The output $\tilde{\mathbf{X}}'$ is written back to the global memory, and the SRAM memory is thus released. Next, we slice the non-reduction dimension of $\tilde{\mathbf{X}}'$ and $\mathbf{P}_2$ with tiling size $b_{n_2}$, and compute the matrix multiplication, followed by quantizing the result on the fly.

**Kernel Profiling.** We enumerate popular hidden sizes in the series of LLaMA models, and provide the detailed profiling results of FLATQUANT's online transformation with and without kernel fusion in Table 9. Note that the SRAM can hold all of these shapes with the default design on the NVIDIA RTX 3090. It can be found that kernel fusion achieves significant speedup across various hidden dimensions and batch sizes, e.g., 1.5x-3x prefill speedup and 1.2x-4x decoding speedup, respectively.

| Hidden Dimension | Batch Size | without Kernel Fusion | | with Kernel Fusion | | Speedup | |
|---|---|---|---|---|---|---|---|
| | | Prefill Time (ms) | Decode Time (ms) | Prefill Time (ms) | Decode Time (ms) | Prefill | Decode |
| 4096 | 1 | 0.1956 | 0.0184 | 0.0625 | 0.0082 | 3.13x | 2.25x |
| | 2 | 0.3809 | 0.0195 | 0.1116 | 0.0072 | 3.41x | 2.71x |
| | 4 | 0.7199 | 0.0212 | 0.2120 | 0.0082 | 3.40x | 2.59x |
| | 8 | 1.4019 | 0.0236 | 0.4188 | 0.0082 | 3.35x | 2.88x |
| | 16 | 2.7628 | 0.0307 | 0.8417 | 0.0073 | 3.28x | 4.20x |
| | 32 | 5.5101 | 0.0317 | 1.7091 | 0.0082 | 3.22x | 3.87x |
| | 64 | 10.9752 | 0.0328 | 3.4898 | 0.0082 | 3.14x | 4.00x |
| 5120 | 1 | 0.2519 | 0.0195 | 0.1321 | 0.0113 | 1.91x | 1.73x |
| | 2 | 0.4915 | 0.0205 | 0.2570 | 0.0113 | 1.91x | 1.82x |
| | 4 | 0.9073 | 0.0225 | 0.5161 | 0.0113 | 1.76x | 2.00x |
| | 8 | 1.7582 | 0.0266 | 1.0363 | 0.0113 | 1.70x | 2.36x |
| | 16 | 3.4748 | 0.0338 | 2.0480 | 0.0121 | 1.70x | 2.80x |
| | 32 | 6.9079 | 0.0358 | 4.1313 | 0.0123 | 1.67x | 2.92x |
| | 64 | 13.8619 | 0.0379 | 8.2033 | 0.0123 | 1.69x | 3.08x |
| 8192 | 1 | 0.3845 | 0.0195 | 0.1608 | 0.0132 | 2.39x | 1.48x |
| | 2 | 0.7393 | 0.0205 | 0.3092 | 0.0132 | 2.39x | 1.55x |
| | 4 | 1.4433 | 0.0205 | 0.6257 | 0.0123 | 2.31x | 1.67x |
| | 8 | 2.8529 | 0.0215 | 1.2411 | 0.0133 | 2.30x | 1.62x |
| | 16 | 5.6668 | 0.0225 | 2.4904 | 0.0133 | 2.28x | 1.69x |
| | 32 | 11.3183 | 0.0246 | 4.9418 | 0.0133 | 2.29x | 1.85x |
| | 64 | 22.6714 | 0.0297 | 9.8459 | 0.0143 | 2.30x | 2.07x |
| 11008 | 1 | 0.6154 | 0.0215 | 0.3830 | 0.0173 | 1.61x | 1.24x |
| | 2 | 1.2032 | 0.0225 | 0.7547 | 0.0173 | 1.59x | 1.30x |
| | 4 | 2.3654 | 0.0223 | 1.5032 | 0.0164 | 1.57x | 1.36x |
| | 8 | 4.7570 | 0.0236 | 2.9983 | 0.0174 | 1.59x | 1.35x |
| | 16 | 9.4536 | 0.0256 | 6.0099 | 0.0184 | 1.57x | 1.39x |
| | 32 | 18.9102 | 0.0287 | 12.0444 | 0.0195 | 1.57x | 1.47x |
| | 64 | 38.2700 | 0.0379 | 24.0000 | 0.0248 | 1.59x | 1.53x |
| 13824 | 1 | 0.7260 | 0.0225 | 0.4444 | 0.0184 | 1.63x | 1.22x |
| | 2 | 1.4203 | 0.0236 | 0.8653 | 0.0184 | 1.64x | 1.28x |
| | 4 | 2.8088 | 0.0246 | 1.7254 | 0.0184 | 1.63x | 1.33x |
| | 8 | 5.6228 | 0.0247 | 3.4273 | 0.0195 | 1.64x | 1.27x |
| | 16 | 11.2297 | 0.0266 | 6.8726 | 0.0195 | 1.63x | 1.37x |
| | 32 | 22.4302 | 0.0319 | 13.7216 | 0.0205 | 1.63x | 1.56x |
| | 64 | 45.4374 | 0.0471 | 27.4698 | 0.0275 | 1.65x | 1.72x |
| 14336 | 1 | 0.6932 | 0.0215 | 0.4178 | 0.0184 | 1.66x | 1.17x |
| | 2 | 1.3466 | 0.0225 | 0.8233 | 0.0184 | 1.64x | 1.22x |
| | 4 | 2.6557 | 0.0236 | 1.6507 | 0.0184 | 1.61x | 1.28x |
| | 8 | 5.2910 | 0.0246 | 3.2922 | 0.0195 | 1.61x | 1.26x |
| | 16 | 10.5185 | 0.0257 | 6.5966 | 0.0195 | 1.59x | 1.32x |
| | 32 | 20.9249 | 0.0317 | 13.0601 | 0.0205 | 1.60x | 1.55x |
| | 64 | 42.7981 | 0.0461 | 25.9308 | 0.0266 | 1.65x | 1.73x |

Table 9: Prefill and decoding speedup of kernel fusion across different hidden dimensions and batch sizes. The sequence length is 2048 for prefill and 1 for decoding. The default kernel design holds for all the above settings.

We also selectively test the two corner cases with the hidden size of 28762, both of which bring considerably 2.3x speedup.

## C  ADDITIONAL EXPERIMENTS

### C.1  FLATQUANT LEADS TO FLATNESS

In Figure 1 and Figure 2, we illustrate that the affine transformations of FlatQuant actually learn to flatten the distribution of weights and activations to lower the quantization error, even though we do not include flatness in the optimization target. The simple layer-wise MSE loss defined in Equation 4 is enough to encourage flatness in weights and activations. In the following, we validate that with more experiment results.

**Quantifying the Flatness.**  We quantify the flatness of weights and activations by calculating the channel distribution (a 1D vector $d$) as Figure 1 and measuring its mean squared error (MSE) against a perfectly flat distribution ($d'$). The flat distribution $d'$ is defined with equal magnitudes across all channels and the same L2 norm as $d$.

**FLATQUANT Leads to Flatness.**  We find that flatness well explains the optimization process of FlatQuant. In Figure 8, we plot the flatness and mean squared quantization error (MSE) of different

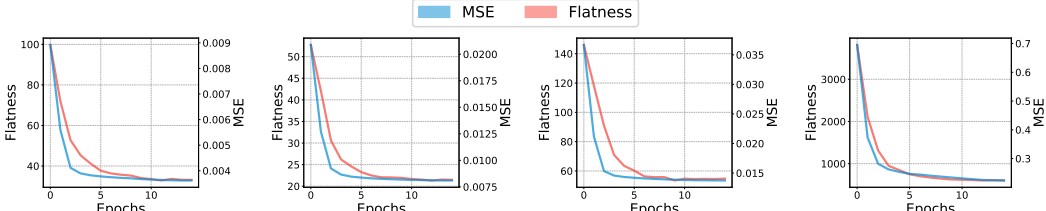

(a) 7$^{th}$ Transformer block. (b) 15$^{th}$ Transformer block. (c) 23$^{th}$ Transformer block. (d) 31$^{th}$ Transformer block.

Figure 8: Flatness and mean squared quantization error (MSE) of different Transformer blocks in LLaMA-3-8B during FLATQUANT's training process. Flatness is calculated as the total flatness of all weights and activations within a Transformer block.

Transformer blocks in LLaMA-3-8B during FLATQUANT's training process. It can be observed that as the training loss decreases, the flatness is also improved, which indicates that FlatQuant learns better transformation to obtain a flatter distribution which eventually contributes to smaller quantization error, i.e. flatness matters for LLM quantization.

## C.2 MORE EXPERIMENTAL RESULTS

**Results on LLaMA-3.1-8B-Instruct.** Besides the results of MT-Bench, we present the PPL results and the performance on QA tasks for LLaMA-3.1-8B-Instruct model in Table 10.

|  | WikiText-2 | C4 | ARC-C | ARC-E | HellaSwag | LAMBADA | PIQA | Winogrande | Avg |
|---|---|---|---|---|---|---|---|---|---|
| FP16 | 7.22 | 11.38 | 55.20 | 79.67 | 79.20 | 73.14 | 81.12 | 73.80 | 73.69 |
| FLATQUANT | 7.97 | 12.99 | 52.90 | 79.25 | 76.68 | 70.79 | 79.49 | 73.09 | 72.03 |

Table 10: Evaluation results of FLATQUANT on LLaMA-3.1-8B-Instruct.

**Results on Qwen-2.5-Instruct.** To further validate the generality of FlatQuant, we conduct experiments on the Qwen-2.5-Instruct models. FlatQuant achieves near-lossless quantization (e.g., only 0.21% accuracy loss on QA tasks for Qwen-2.5-Instruct-32B). The results on language modeling and QA benchmarks are summarized in Table 11.

| Model | Method | W Quantizer | WikiText-2 | C4 | ARC-C | ARC-E | HellaSwag | LAMBADA | PIQA | Winogrande | Avg |
|---|---|---|---|---|---|---|---|---|---|---|---|
| **7B** | FP16 | - | 8.36 | 14.37 | 51.37 | 75.80 | 79.57 | 67.61 | 80.20 | 69.93 | 70.75 |
|  | FLATQUANT | RTN | 8.46 | 13.94 | 51.71 | 77.69 | 78.42 | 57.46 | 76.93 | 69.53 | 68.62 |
|  | FP16 | - | 5.32 | 10.45 | 58.62 | 77.02 | 85.25 | 75.14 | 81.39 | 73.16 | 75.10 |
| **32B** | QuaRot | RTN | 6.95 | 12.17 | 52.13 | 74.37 | 80.41 | 68.37 | 78.45 | 67.72 | 70.24 |
|  | QuaRot | GPTQ | 6.54 | 11.65 | 56.06 | 76.52 | 81.83 | 71.26 | 78.78 | 69.06 | 72.25 |
|  | FLATQUANT | RTN | 5.80 | 10.86 | 58.62 | 78.58 | 83.72 | 75.26 | 80.74 | 72.45 | 74.89 |

Table 11: Evaluation results of FLATQUANT on Qwen-2.5-Instruct models.

**KV Cache Quantization.** As introduced in Section 4.4, while our primary focus is on weight-activation quantization, FLATQUANT serves as a general framework applicable to various quantization tasks. To further evaluate its versatility, we apply FLATQUANT to KV cache only quantization. In this setting, we retain high precision for the rest of the model (including weights and activations) and apply the group-wise asymmetric quantization (with a group size of 128) to keys and values. Table 12 presents the results of KV cache quantization using various bit-widths on the LLaMA-3-8B model. Consistent with previous studies (Hooper et al., 2024; Liu et al., 2024c; Ashkboos et al., 2024), we observe that keys are more sensitive to quantization than values. Furthermore, Table 13 compares FLATQUANT with QuaRot for KV cache quantization on LLaMA-2-7B and LLaMA-2-13B models. As shown, FLATQUANT delivers superior performance in most cases, particularly for lower-bit (2-3 bits). When both keys and values are quantized to 2 bits, FLATQUANT outperforms QuaRot by 2.57 in perplexity for the 7B model.

| K bits | V bits | WikiText-2 | C4 | ARC-C | ARC-E | HellaSwag | LAMBADA | PIQA | Winogrande | Avg |
|---|---|---|---|---|---|---|---|---|---|---|
| 16 | 16 | 6.14 | 9.45 | 53.50 | 77.57 | 79.12 | 75.51 | 80.74 | 72.93 | 73.23 |
| 4 | 4 | 6.20 | 9.56 | 52.82 | 78.20 | 79.13 | 75.32 | 80.47 | 72.77 | 73.12 |
| 4 | 3 | 6.25 | 9.66 | 52.90 | 77.65 | 79.00 | 75.10 | 80.79 | 73.48 | 73.15 |
| 4 | 2 | 6.60 | 10.33 | 49.32 | 74.37 | 77.88 | 72.77 | 79.22 | 72.69 | 71.04 |
| 3 | 4 | 6.35 | 9.91 | 52.05 | 77.95 | 78.41 | 73.94 | 79.71 | 73.48 | 72.59 |
| 3 | 3 | 6.41 | 10.03 | 52.47 | 76.85 | 78.25 | 74.02 | 79.98 | 72.61 | 72.36 |
| 3 | 2 | 6.84 | 10.83 | 47.44 | 73.91 | 77.18 | 70.37 | 78.73 | 71.19 | 69.80 |
| 2 | 4 | 7.70 | 13.36 | 49.15 | 74.62 | 74.74 | 63.65 | 77.58 | 68.67 | 68.07 |
| 2 | 3 | 7.79 | 13.44 | 46.67 | 71.63 | 74.17 | 63.05 | 77.48 | 68.51 | 66.92 |
| 2 | 2 | 8.93 | 16.13 | 42.92 | 68.60 | 71.54 | 55.58 | 75.30 | 64.40 | 63.06 |

Table 12: Different bits for KV cache quantization on the LLaMA-3-8B model.

| Methods | K bits | V bits | LLaMA-2-7B | LLaMA-2-13B |
|---|---|---|---|---|
| | 16 | 16 | 5.47 | 4.88 |
| QuaRot | 4 | 4 | 5.51 | 4.91 |
| | 3 | 3 | 5.68 | 5.02 |
| | 2 | 2 | 9.23 | 7.07 |
| FLATQUANT | 4 | 4 | 5.50 | 4.91 |
| | 3 | 3 | 5.61 | 5.00 |
| | 2 | 2 | 6.66 | 5.69 |

Table 13: WikiText-2 perplexity of LLaMA-2 models with different bits of KV cache quantization.

**Extreme Low-bit Quantization.** We quantize the LLM to extreme low-bit representations (e.g., INT3) to investigate the limitations of quantization. The results in Table 14 show that FLATQUANT still keeps most of the model's abilities in the 3-bit setting, whereas QuaRot struggles under such extreme low-bit conditions. Nevertheless, 4-bit quantization remains a better balance between inference resource efficiency and acceptable performance degradation for now.

## C.3 ADDITIONAL DISCUSSIONS

**Calibration Set.** Since FLATQUANT employs a gradient-based method to optimize transformations for increased flatness, one reasonable concern is whether FLATQUANT might overfit the calibration set. To assess its generalization ability, we conducted an ablation study using different calibration datasets: WikiText-2, C4, and Pile. As shown in Table 15, FLATQUANT maintains stable performance across all datasets. For example, when calibrated on different datasets, FLATQUANT exhibits similar performance on WikiText-2, with PPL ranging from 6.98 to 7.04. On the C4 dataset, results are equally consistent, with PPLs between 11.05 and 11.13. Furthermore, QA accuracy remains within a narrow range (71.04% to 71.23%), suggesting that FLATQUANT generalizes well across different calibration datasets. This robustness is attributed to FLATQUANT's focus on learning an equivalent affine transformation with minimal quantization loss, rather than altering the model's weights. Nevertheless, it is reasonable to assume that the diversity of calibration data can further enhance the performance of our method.

**Effect of Clipping.** Unlike weight clipping, which has been widely utilized in LLM quantization, activation clipping has been less explored. Although previous studies (Ashkboos et al., 2024; Liu et al., 2024b) show that activation clipping offers only modest benefits for quantization, our

| LLaMA3-8B | WikiText-2 | C4 | ARC-C | ARC-E | HellaSwag | LAMBADA | PIQA | Winogrande | Avg |
|---|---|---|---|---|---|---|---|---|---|
| FP16 | 6.14 | 9.45 | 53.50 | 77.57 | 79.12 | 75.51 | 80.74 | 72.93 | 73.23 |
| QuaRot-W4A4KV4 | 8.16 | 13.38 | 45.73 | 70.83 | 72.97 | 62.70 | 75.35 | 67.17 | 65.79 |
| FLATQUANT-W4A4KV4 | 6.98 | 11.13 | 50.00 | 75.80 | 76.80 | 72.91 | 79.16 | 72.69 | 71.23 |
| QuaRot-W3A3KV3 | 686.54 | 630.89 | 25.34 | 28.41 | 28.07 | 0.78 | 50.71 | 48.70 | 30.33 |
| FLATQUANT-W3A3KV3 | 10.82 | 19.03 | 35.41 | 63.26 | 65.30 | 52.49 | 73.56 | 60.69 | 58.45 |

Table 14: Extreme low bit quantization results on LLAMA-3-8B models.

| Calibration set | WikiText-2 | C4 | ARC-C | ARC-E | HellaSwag | LAMBADA | PIQA | Winogrande | Avg |
|---|---|---|---|---|---|---|---|---|---|
| WikiText2 | 6.98 | 11.13 | 50.00 | 75.80 | 76.80 | 72.91 | 79.16 | 72.69 | 71.23 |
| C4 | 7.04 | 11.05 | 50.34 | 75.38 | 76.74 | 73.28 | 78.67 | 71.82 | 71.04 |
| Pile | 7.04 | 11.08 | 51.11 | 77.36 | 76.63 | 72.37 | 78.94 | 70.56 | 71.16 |

Table 15: Ablation study of FLATQUANT's calibration set on LLaMA-3-8B model.

method demonstrates that LCT provides significant improvements. As shown in Table 16, applying our transformations prior to clipping allows for a greater proportion of values to be clipped, resulting in better performance. In contrast, applying LCT before the transformation, similar to the approach used in RTN quantization, yields only limited improvements. This is consistent with prior findings (Dettmers et al., 2022) and is largely due to the presence of severe outliers in activation. We also report results using a QuaRot-style clipping method, with 0.9 as the activation clipping threshold and 0.95 for the KV cache clipping threshold. In summary, the integration of transformations enhances the effectiveness of clipping, indicating that their combination significantly improves weight-activation quantization.

| LLaMA3-8B | WikiText-2 | C4 | ARC-C | ARC-E | HellaSwag | LAMBADA | PIQA | Winogrande | Avg |
|---|---|---|---|---|---|---|---|---|---|
| FP16 | 6.14 | 9.45 | 53.50 | 77.57 | 79.12 | 75.51 | 80.74 | 72.93 | 73.23 |
| w/o LCT | 7.95 | 12.74 | 44.20 | 71.89 | 74.21 | 68.72 | 77.15 | 66.30 | 67.08 |
| LCT before Transformation | 7.37 | 11.86 | 48.72 | 76.18 | 75.11 | 66.65 | 77.91 | 67.17 | 68.62 |
| QuaRot-style Fixed Threshold | 7.25 | 11.62 | 48.21 | 75.29 | 75.66 | 71.32 | 78.73 | 70.01 | 69.87 |
| LCT after Transformation | 6.98 | 11.13 | 50.00 | 75.80 | 76.80 | 72.91 | 79.16 | 72.69 | 71.23 |

Table 16: The effect of Learnable Clipping Thresholds.

## C.4 EXPERIMENT DETAILS OF FIGURE 5

In Figure 5, we present the prefill speedup and WikiText2 PPL results of different decomposed matrix sizes on LLaMA-2-7B model. We decompose the hidden dimension 4096 into $n_1 \times n_2$ and range $n_1$ from 1 to 2048, where $n_1 = 1$ amounts to maintaining a full-size transformation matrix. The intermediate dimension 11008 is decomposed into $64 \times 172$ as done in FLATQUANT. For PPL evaluation, we only quantize the last Transformer block and learn the affine transformations within it. For speedup evaluation, we do not leverage the online transformation kernel in Section 3.3 and implement online transformations with naive matrix multiplication in PyTorch.

## C.5 ADDITIONAL ANALYSES OF INFERENCE LATENCY

**Baseline.** We implement and report the latency results of INT4 quantization and QuaRot with QuaRot's official code[4]. These baselines share the same quantization settings with FLATQUANT as described in Section 4.1 for fair comparison.

**End-to-end Speedup.** We decode 256 tokens after the prefill on a sequence length of 2048 and provide the prefill and decoding speedup of FLATQUANT in Figure 9 and Figure 10. FLATQUANT achieves a prefill speedup of 2.30x and decoding speedup of 1.76x under the batch size of 64, with only 0.07x speedup loss compared to the naive INT4 quantization for both prefill and decoding. Note that when the batch size is smaller than 16, quantization overhead outweighs the benefits brought by KV cache memory reduction for the decoding stage, resulting in less than 1x speedup for both INT4 quantization and FLATQUANT. However, since the decoding speedup shows good scalability with the batch size, we can gain a practical decoding speedup simply by employing a large batch size.

**Total FLOPs of Online Transformations.** (1) Self-Attention. The self-attention module has three online transformations, i.e., $\{\mathbf{P}_a, \mathbf{P}_o, \mathbf{P}_h\}$. Suppose the hidden dimension $h_d$ and intermediate dimension $h_i$ of LLM can be perfectly decomposed into $\sqrt{h_d} \times \sqrt{h_d}$ and $\sqrt{h_i} \times \sqrt{h_i}$, respectively, then the total FLOPs of $\{\mathbf{P}_a, \mathbf{P}_o, \mathbf{P}_h\}$ is $4bsh_d\sqrt{h_d} + 2bsh_d a + 4bsh_d^2/a$, where $b$ is the batch size, $s$

---

[4]https://github.com/spcl/QuaRot

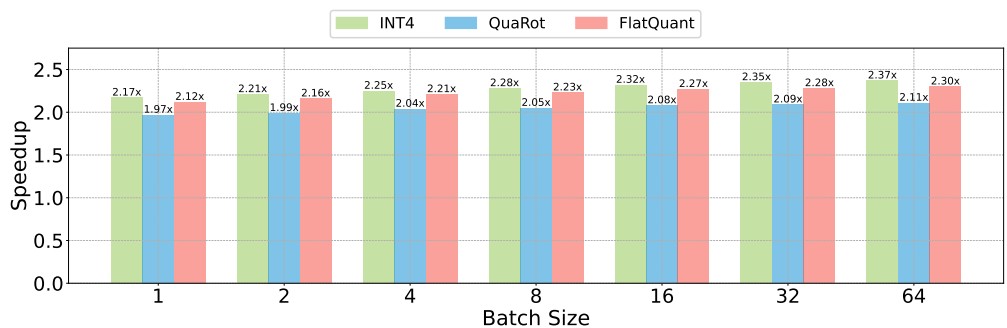

Figure 9: Prefill speedup of LLaMA-2-7B on a sequence length of 2048.

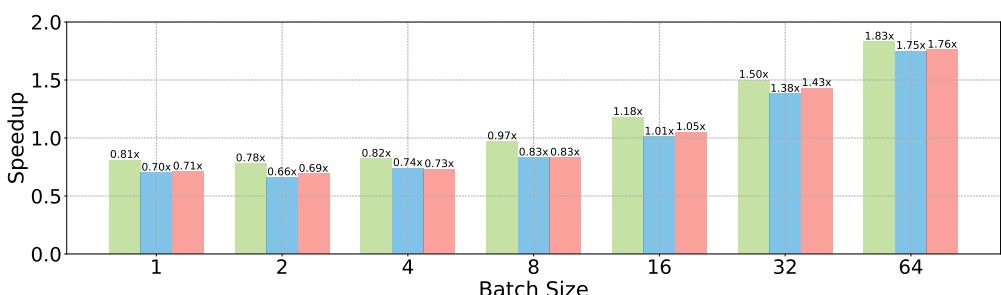

Figure 10: Decoding speedup on LLaMA-2-7B model. We decode 256 tokens after the prefill on a sequence length of 2048.

is the sequence length, and $a$ is the number of attention heads. (2) Feed-forward Network. The feed-forward module has two online transformations, i.e., $\{\mathbf{P}_{ug}, \mathbf{P}_d\}$. The total FLOPs of $\{\mathbf{P}_{ug}, \mathbf{P}_d\}$ is $4bsh_d\sqrt{h_d} + 4bsh_i\sqrt{h_i}$. In summary, the total FLOPs of the online transformations in a Transformer block amounts to $8bsh_d\sqrt{h_d} + 2bsh_da + 4bsh_d^2/a + 4bsh_i\sqrt{h_i}$. In LLaMA-2-7B (i.e., $h_d = 4096$, $h_i = 11008$ and $a = 32$), the FLOPs of online transformations only account for about $2.61\%$ of those of the FP16 model when $s$ reaches 2048.

**Memory Consumption of Online Transformations.** We compute the parameter count of each online transformation below: (1) $\mathbf{P}_a$: $2(\sqrt{h_d})^2$; (2) $\mathbf{P}_o$: $a^2$; (3) $\mathbf{P}_h$: $(h_d/a)^2$; (4) $\mathbf{P}_{ug}$: $2(\sqrt{h_d})^2$; (5) $\mathbf{P}_d$: $2(\sqrt{h_i})^2$. The total parameter count in one Transformer block is $4h_d + 2h_i + a^2 + (h_d/a)^2$. The additional memory consumption during inference is $2(4h_d + 2h_i + a^2 + (h_d/a)^2)$ bytes, which only consumes about 0.11MB extra memory space for LLaMA-2-7B.

# D   ADDITIONAL VISUALIZATIONS

## D.1   MORE VISUALIZATIONS OF WEIGHT AND ACTIVATION DISTRIBUTIONS

**Experiment Details.** We visualize the distribution of weights and activations after different transformations, including per-channel scaling in SmoothQuant (Xiao et al., 2023), Hadamard transformation in QuaRot (Ashkboos et al., 2024), and affine transformation in FLATQUANT. We compute the per-channel Frobenius norm to quantify the channel magnitude. We randomly sample from the C4 (Raffel et al., 2020) dataset to collect activation statistics.

**Visualizations on the LLaMA Models.** We visualize the distribution envelopes of both original and transformed weights and activations on the LLaMA models in Figure 11-17. It can be observed that neither per-channel scaling nor Hadamard transformation can fully smooth out outlier channels to produce flatness, still leaving outlier channels, especially on activations. On the other hand, the affine transformation learned by FLATQUANT can effectively produce flatter distributions for both weights and activations which are easier to quantize.

### D.2    MORE VISUALIZATIONS OF QUANTIZATION ERROR LANDSCAPES

**Experiment Details.**    We randomly sample 128 samples from the C4 (Raffel et al., 2020) dataset and compute their average mean squared error for visualization. For per-channel scaling, we follow SmoothQuant (Xiao et al., 2023) and only perform per-channel scaling for the inputs of the self-attention and feed-forward modules. For the Hadamard transformation, we replace the affine transformation in FLATQUANT with a fixed Hadamard transformation. The quantization settings are the same as those described in Section 4.1.

**Visualizations on the LLaMA Models.**    We visualize the quantization error landscapes of LLaMA models in Figure 2 and Figure 18-21. With the affine transformation to smooth outliers, FLATQUANT can effectively suppress the quantization errors at pivot tokens and ease the quantization error propagation, leading to a flatter quantization error landscape compared with per-channel scaling and Hadamard transformation.

## E    LIMITATIONS

In this study, we present FLATQUANT, but there are certain limitations to acknowledge. First, the full potential of 4-bit quantization has not been thoroughly explored. While we follow the previous studies to build the calibration set and demonstrate that FLATQUANT is robust across various data sources, the optimal selection of calibration sets remains an open question. Additionally, our focus has primarily been on the INT4 data type, and we have not examined the integration of FLATQUANT with newer data types, such as MXFP4, which may offer advantages over INT4. Addressing these aspects represents promising avenues for future research.

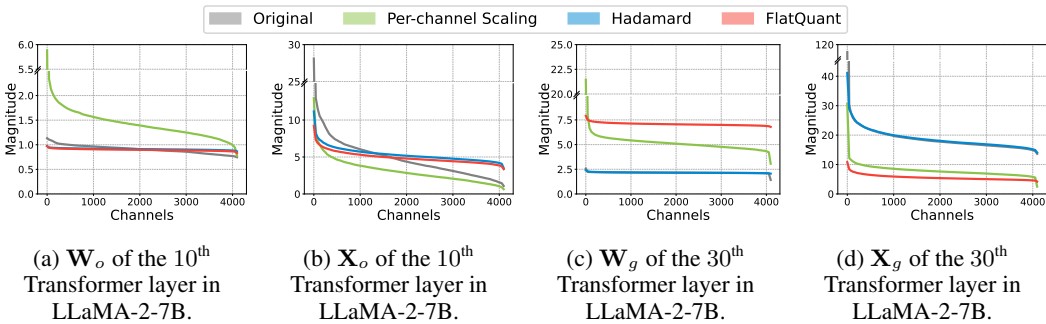

(a) $\mathbf{W}_o$ of the 10th Transformer layer in LLaMA-2-7B.

(b) $\mathbf{X}_o$ of the 10th Transformer layer in LLaMA-2-7B.

(c) $\mathbf{W}_g$ of the 30th Transformer layer in LLaMA-2-7B.

(d) $\mathbf{X}_g$ of the 30th Transformer layer in LLaMA-2-7B.

Figure 11: Distributions of weights and inputs from LLaMA-2-7B, sorted by the channel magnitudes (i.e., the Frobenius norm) in descending order.

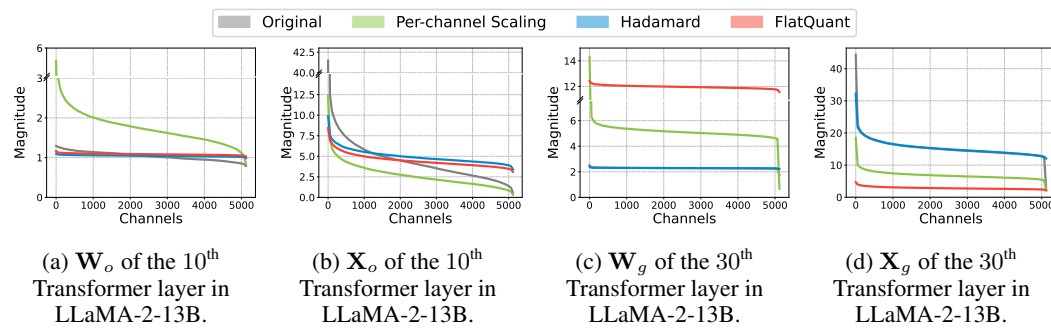

Figure 12: Distributions of weights and inputs from LLaMA-2-13B, sorted by the channel magnitudes (i.e., the Frobenius norm) in descending order.

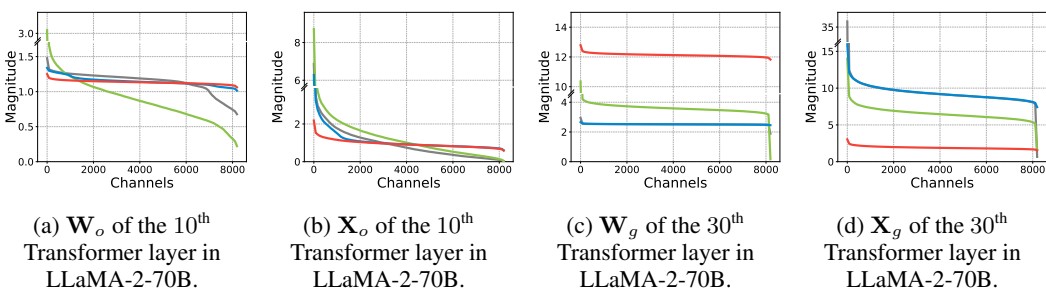

Figure 13: Distributions of weights and inputs from LLaMA-2-70B, sorted by the channel magnitudes (i.e., the Frobenius norm) in descending order.

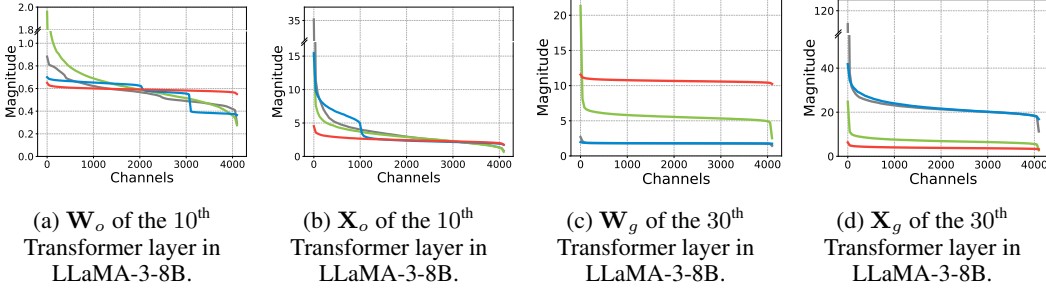

Figure 14: Distributions of weights and inputs from LLaMA-3-8B, sorted by the channel magnitudes (i.e., the Frobenius norm) in descending order.

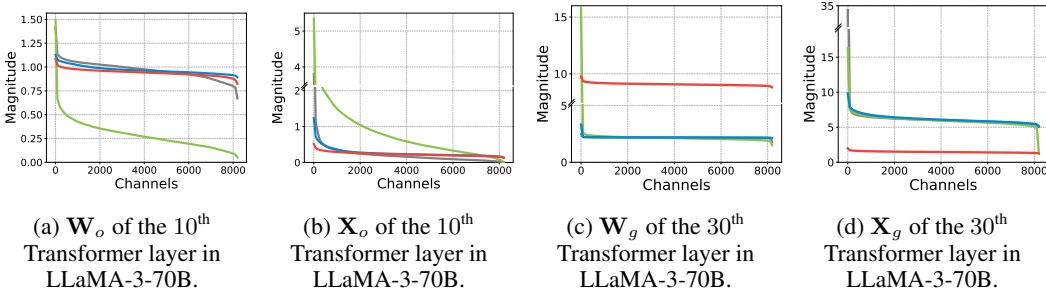

Figure 15: Distributions of weights and inputs from LLaMA-3-70B, sorted by the channel magnitudes (i.e., the Frobenius norm) in descending order.

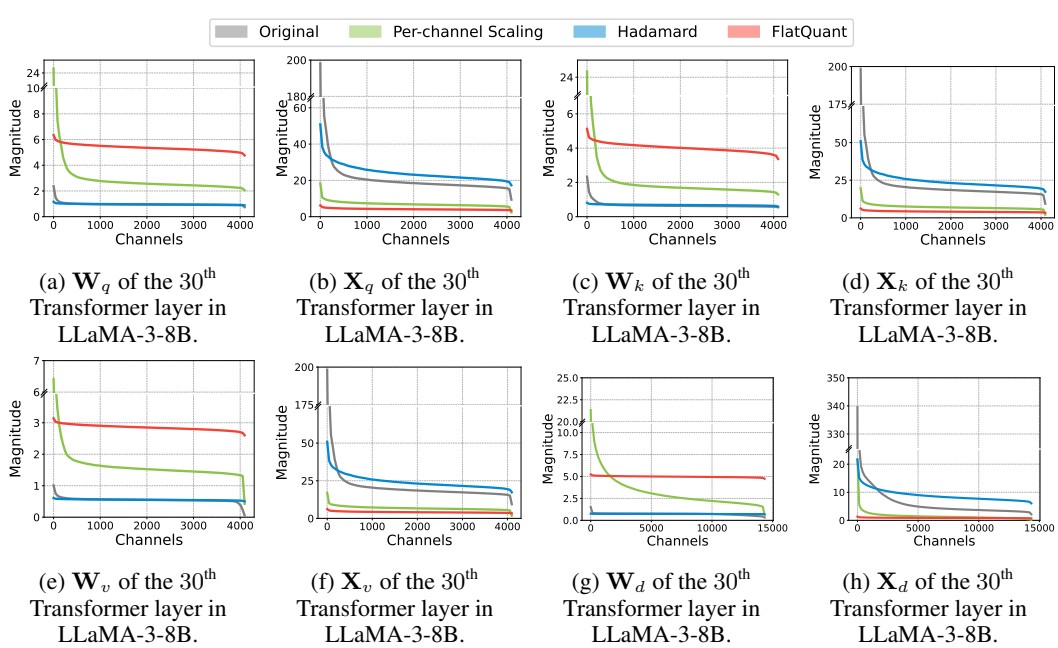

Figure 16: Distributions of weights and inputs from LLaMA-3-8B, sorted by the channel magnitudes (i.e., the Frobenius norm) in descending order.

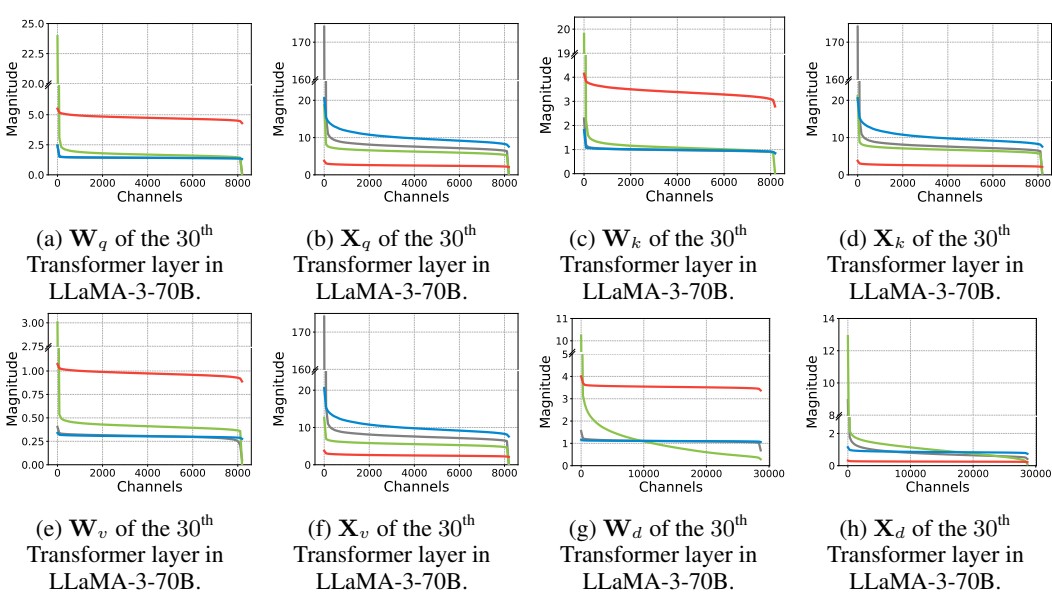

Figure 17: Distributions of weights and inputs from LLaMA-3-70B, sorted by the channel magnitudes (i.e., the Frobenius norm) in descending order.

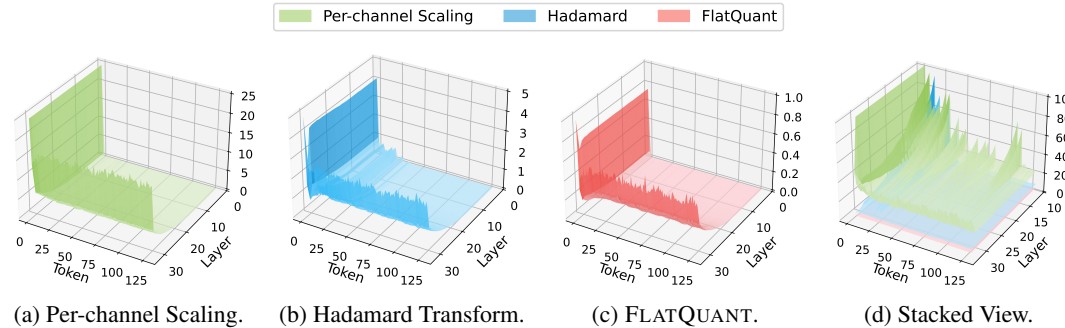

(a) Per-channel Scaling.    (b) Hadamard Transform.    (c) FLATQUANT.    (d) Stacked View.

Figure 18: The mean squared error (MSE) of quantization across Transformer layers and input sequence in LLaMA-2-7B. Figure 18a-18c plot the MSE surface of each method, while Figure 18d overlays these surfaces by dividing each MSE with that of FLATQUANT.

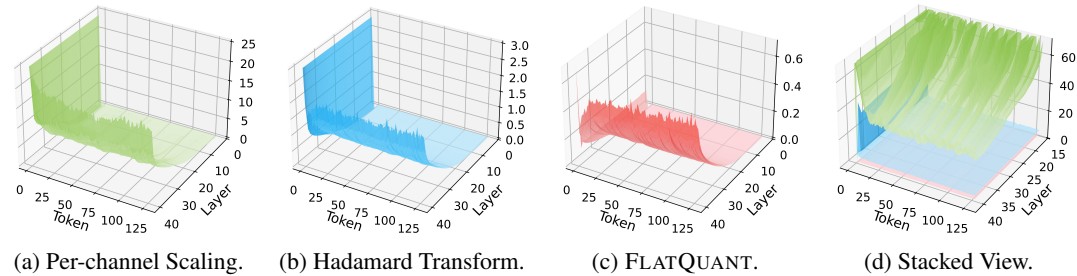

(a) Per-channel Scaling.    (b) Hadamard Transform.    (c) FLATQUANT.    (d) Stacked View.

Figure 19: The mean squared error (MSE) of quantization across Transformer layers and input sequence in LLaMA-2-13B. Figure 19a-19c plot the MSE surface of each method, while Figure 19d overlays these surfaces by dividing each MSE with that of FLATQUANT.

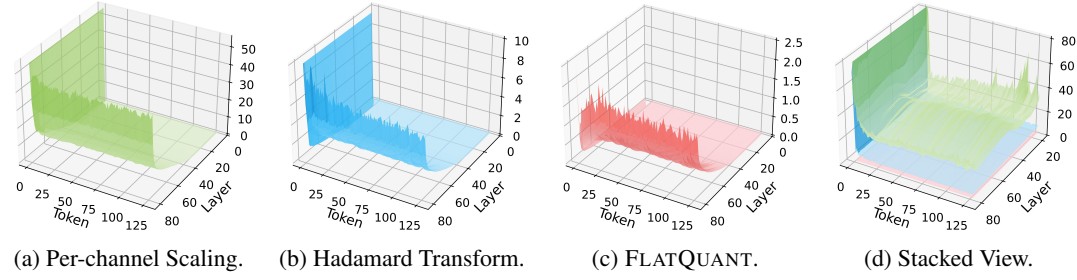

(a) Per-channel Scaling.    (b) Hadamard Transform.    (c) FLATQUANT.    (d) Stacked View.

Figure 20: The mean squared error (MSE) of quantization across Transformer layers and input sequence in LLaMA-2-70B. Figure 20a-20c plot the MSE surface of each method, while Figure 20d overlays these surfaces by dividing each MSE with that of FLATQUANT.

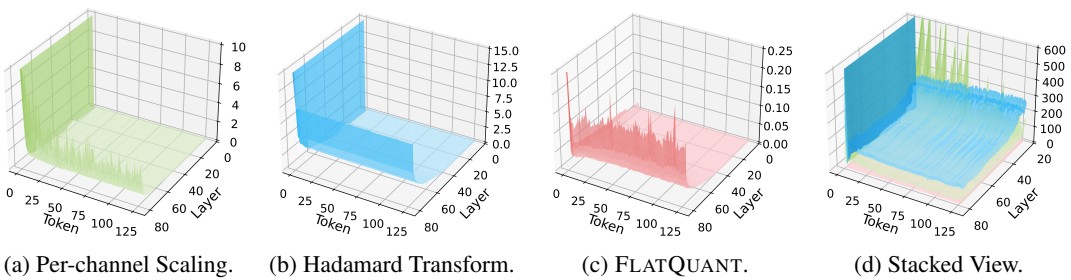

(a) Per-channel Scaling.    (b) Hadamard Transform.    (c) FLATQUANT.    (d) Stacked View.

Figure 21: The mean squared error (MSE) of quantization across Transformer layers and input sequence in LLaMA-3-70B. Figure 21a-21c plot the MSE surface of each method, while Figure 21d overlays these surfaces by dividing each MSE with that of FLATQUANT.

