# OpenReview forum: "FlatQuant: Flatness Matters for LLM Quantization"
_ICLR.cc/2025/Conference — Submitted to ICLR 2025_

### Official Review · Reviewer_tWUe · 2024-10-16

**Soundness:** 3
**Presentation:** 4
**Contribution:** 3
**Rating:** 5
**Confidence:** 5

**Summary:**

FLATQUANT, a new post-training quantization method, enhances weight and activation flatness via optimal affine transformations, calibrated quickly with minimal runtime overhead using Kronecker decomposition. It sets a new state-of-the-art with W4A4 quantization and significantly improves inference speed.

**Strengths:**

1. This paper sets a new state-of-the-art performance.
2. Extensive experiments validate its effectiveness.
3. Fast CUDA kernel implementation enhances its applicability.

**Weaknesses:**

1. The manuscript does not clearly explain how the invertibility of $\mathbf{P}$ is maintained during training. Detailing the methods used to ensure this could improve understanding of the method.
2. I notice the use of Kronecker decomposition during inference and SVD during calibration, but the specific methodologies for Kronecker decomposition are not detailed. Elaborating on these could clarify the approach.
3. The potential approximation errors from Kronecker decomposition and their impact on model performance are not mentioned. Experimental results in this area would be beneficial.
4. The overhead of dynamic activation clipping during inference and the individual contributions of $\alpha_a$ and $\alpha_w$, should be explored.
5. The ablation study should include separate evaluations for "PS", "LCT", and "PS+LCT" to comprehensively demonstrate the improvements from each component.

**Questions:**

1. The distribution of $\mathbf{W}_d$, $\mathbf{W}_q$, $\mathbf{W}_k$, and $\mathbf{W}_v$ and their corresponding activations is not shown. Visualizations like Fig. 2 for these weights and activations would be beneficial.
2. I am curious about the speedup ratio of the method on Hopper architecture GPUs (e.g., H100, H800).
3. Could the authors explain why the per-channel scaling is employed? A proper $\mathbf{P}'$ can contain the ability to balance outliers between the weights and activations, for example,  $\mathbf{P}'=\mathbf{P}\text{dig}(\mathbf{c})$.

---

> ### Author Response · Authors · 2024-11-23
> **Response to Reviewer tWUe (part 1)**
>
> We thank the reviewer for the positive review and constructive comments. In the following, we elaborate on the method part of FlatQuant (including the invertibility of $\mathbf P$ and the use of Kronecker decomposition) and provide more experiment results to showcase the effectiveness of FlatQuant.
>
>
> ---
> **Q1**. *The manuscript does not clearly explain how the invertibility of $\mathbf P$ is maintained during training. Detailing the methods used to ensure this could improve understanding of the method.*
>
> **A1**. We thank the reviewer for raising this important question. As discussed in Appendix B.1, FlatQuant ensures invertibility by employing a singular value decomposition (SVD) re-parameterization trick during training. In the following, we provide more details.
>
> * **The SVD Re-parameterization**. For a learnable affine transformation matrix $\mathbf P$, we decompose it as $\mathbf P=\mathbf U\Sigma\mathbf V^\top$, where  $\mathbf U,\mathbf V$ are orthogonal matrices, and $\Sigma$ is a diagonal matrix. We ensure the invertibility of $\mathbf P$ by forcing the elements of $\Sigma$ larger than zero. The optimization of orthogonal matrices can be readily accomplished with Cayley parameterization. Therefore, $\mathbf P^{-1}$ can be easily computed with $\mathbf V\Sigma^{-1}\mathbf U^\top$.
>
> * **Comparison with Direct Inversion**. In Appendix B.1, we also compare this SVD training strategy with the naive strategy of directly inverting the matrix $\mathbf P$, which cannot ensure the invertibility of $\mathbf P$ theoretically. Although we empirically find that this naive strategy can still produce invertible matrices in the training process, it needs FP32 training to stabilize the optimization process. After thorough comparisons between these two strategies, we finally adopt the SVD training which enjoys both training stability and efficiency.
>
>
> ---
> **Q2**. *I notice the use of Kronecker decomposition during inference and SVD during calibration, but the specific methodologies for Kronecker decomposition are not detailed. Elaborating on these could clarify the approach.*
>
>
> **A2**. We would like to clarify that FlatQuant directly learns the Kronecker-decomposed transformations during calibration and uses them during inference. The detailed process is as follows:
>
> 1. We prepare two individual learnable decomposed matrices of size $n_1$ and $n_2$ for each affine transformation, where $n_1$ and $n_2$ are determined based on the strategy introduced in Section 3.1 (Line 230~234).
>
> 2. Each decomposed matrix is then constructed using SVD re-parameterization trick (detailed in Appendix B.1) to ensure training stability and efficiency, where two orthogonal matrices and one diagonal matrix are used to construct one decomposed matrix.
>
> 3. These transformation matrices are inserted into the network and trained using the calibration data.
>
> 4. After training, the SVD matrices can be combined to obtain the learned decomposed transformation matrices. After that, as shown in Figure 3, some of the transformation matrices, along with other learnable parameters, are merged into the weights. The remaining transformations are retained online and efficiently executed using our fused kernel.
>
>
> ---
> **Q3**. *The potential approximation errors from Kronecker decomposition and their impact on model performance are not mentioned. Experimental results in this area would be beneficial.*
>
> **A3**. We thank the reviewer for the advice. Our experiments demonstrate that Kronecker decomposition achieves comparable accuracy to full-size transformations while significantly improving inference efficiency.
>
> * As shown in Figure 5, different decomposition sizes have minimal impact on perplexity. However, the choice of decomposition size significantly affects inference efficiency.
>
> * To further validate the impact of decomposition on accuracy, we conducted experiments on LLaMA-3-8B with and without decomposition. The table below shows that decomposition maintains high accuracy:
>
> | **Method**        | **WikiText-2** | **C4** | **ARC-C** | **ARC-E** | **HellaSwag** | **LAMBADA** | **PIQA** | **Winogrande** | **Avg.** |
> | :---------------- | :------------- | :----- | :-------- | :-------- | :------------ | :---------- | :------- | :------------- | :------- |
> | w/o Decomposition | 6.88           | 11.23  | 48.98     | 76.77     | 75.92         | 73.61       | 79.54    | 71.11          | 70.99    |
> | w/ Decomposition  | 6.98           | 11.13  | 50.00     | 75.80     | 76.80         | 72.91       | 79.16    | 72.69          | 71.23    |

---

> ### Author Response · Authors · 2024-11-23
> **Response to Reviewer tWUe (part 2)**
>
> **Q4**. *The overhead of dynamic activation clipping during inference should be explored.*
>
>
> **A4**. We thank the reviewer for the suggestion. We measure the latency of quantizing activations of different shapes both with and without activation clipping. The results are shown in the following table.
>
> * As can be seen, activation clipping incurs marginal latency overhead since it only amounts to adding a scaler scaling factor to the calculated max and min tensor with shape [bsz, seq_len].
>
> | **Activation Shape** | **Quantizer w/o Activation Clipping** | **Quantizer w/ Activation Clipping** |
> | :------------------- | :------------------------------------ | :----------------------------------- |
> | [1, 1, 4096]         | 0.144                                 | 0.130                                |
> | [1, 512, 4096]       | 0.114                                 | 0.127                                |
> | [1, 1024, 4096]      | 0.125                                 | 0.156                                |
> | [1, 2048, 4096]      | 0.092                                 | 0.119                                |
> | [16, 1, 4096]        | 0.131                                 | 0.141                                |
> | [16, 512, 4096]      | 0.308                                 | 0.311                                |
> | [16, 1024, 4096]     | 0.583                                 | 0.582                                |
> | [16, 2048, 4096]     | 1.128                                 | 1.132                                |
> | [32, 1, 4096]        | 0.133                                 | 0.156                                |
> | [32, 512, 4096]      | 0.585                                 | 0.583                                |
> | [32, 1024, 4096]     | 1.129                                 | 1.131                                |
> | [32, 2048, 4096]     | 2.227                                 | 2.233                                |
> | [64, 1, 4096]        | 0.153                                 | 0.178                                |
> | [64, 512, 4096]      | 1.130                                 | 1.130                                |
> | [64, 1024, 4096]     | 2.230                                 | 2.233                                |
> | [64, 2048, 4096]     | 4.437                                 | 4.437                                |
>
>
> ---
> **Q5**. *The individual contributions of $\alpha_a$ and $\alpha_w$ should be explored.*
>
> **A5**. We appreciate the reviewer's advice. We conduct additional ablation studies for learnable weight clipping and learnable activation clipping as a complement to Table 4. As shown in the table below, both learnable weight clipping and activation clipping are effective components to increase quantization accuracy.
>
> | **LLaMA3-8B**                   | **WikiText-2** | **C4** | **ARC-C** | **ARC-E** | **HellaSwag** | **LAMBADA** | **PIQA** | **Winogrande** | **Avg.** |
> | :------------------------------ | :------------- | :----- | :-------- | :-------- | :------------ | :---------- | :------- | :------------- | :------- |
> | FP16                            | 6.14           | 9.45   | 53.50     | 77.57     | 79.12         | 75.51       | 80.74    | 72.93          | 73.23    |
> | FlatQuant w/o LCT               | 7.95           | 12.74  | 44.20     | 71.89     | 74.21         | 68.72       | 77.15    | 66.30          | 67.08    |
> | + Learnable Weight Clipping     | 7.43           | 11.91  | 47.10     | 73.19     | 75.35         | 70.54       | 77.04    | 69.06          | 68.71    |
> | + Learnable Activation Clipping | 6.98           | 11.13  | 50.00     | 75.80     | 76.80         | 72.91       | 79.16    | 72.69          | 71.23    |

---

> ### Author Response · Authors · 2024-11-23
> **Response to Reviewer tWUe (part 3)**
>
> **Q6**. *The ablation study should include separate evaluations for "PS", "LCT", and "PS+LCT" to comprehensively demonstrate the improvements from each component.*
>
> **A6**. We thank the reviewer for pointing this out. We include ablation results for "PS", "LCT", and "PS+LCT" in the following table as a complement to Table 4. It is shown that without the online transformations to smooth outliers, the quantized models completely collapse.
>
> | **Method** | **WikiText-2** | **C4**  | **ARC-C** | **ARC-E** | **HellaSwag** | **LAMBADA** | **PIQA** | **Winogrande** | **Avg.** |
> | :--------- | :------------- | :------ | :-------- | :-------- | :------------ | :---------- | :------- | :------------- | :------- |
> | FP16       | 6.14           | 9.45    | 53.50     | 77.57     | 79.12         | 75.51       | 80.74    | 72.93          | 73.23    |
> | PS         | NaN            | NaN     | 22.70     | 25.08     | 25.04         | 0.00        | 49.51    | 49.57          | 28.65    |
> | LCT        | 1149.08        | 1490.08 | 22.95     | 29.29     | 27.35         | 0.60        | 52.99    | 50.83          | 30.67    |
> | PS+LCT     | 8197.96        | 4654.07 | 25.43     | 25.72     | 25.96         | 71.20       | 78.13    | 71.35          | 69.81    |
>
> We further build "PS" and "LCT" on top of online Random Orthogonal Transformations (ROT) to better showcase the strength of "PS" and "LCT". It can be seen that each component contributes positively to the final model accuracy.
>
> | **Method** | **WikiText-2** | **C4** | **ARC-C** | **ARC-E** | **HellaSwag** | **LAMBADA** | **PIQA** | **Winogrande** | **Avg.** |
> | :--------- | :------------- | :----- | :-------- | :-------- | :------------ | :---------- | :------- | :------------- | :------- |
> | FP16       | 6.14           | 9.45   | 53.50     | 77.57     | 79.12         | 75.51       | 80.74    | 72.93          | 73.23    |
> | ROT        | 1266.60        | 936.41 | 25.26     | 28.62     | 27.04         | 1.26        | 51.80    | 51.93          | 30.99    |
> | +PS        | 12.11          | 24.57  | 32.76     | 50.04     | 63.36         | 47.86       | 65.89    | 60.46          | 53.39    |
> | +LCT       | 10.99          | 19.52  | 37.97     | 59.51     | 66.11         | 57.21       | 71.22    | 65.19          | 59.53    |
> | +PS+LCT    | 7.28           | 11.70  | 48.29     | 73.78     | 76.12         | 71.20       | 78.13    | 71.35          | 69.81    |
>
>
> ---
> **Q7**. *The distribution of $\mathbf W_d$, $\mathbf W_q$, $\mathbf W_k$, and $\mathbf W_v$ and their corresponding activations is not shown. Visualizations like Fig. 2 for these weights and activations would be beneficial.*
>
> **A7**. We thank the reviewer for the constructive advice. We have added visualizations for the weights and activations of these linear layers in Figure 16-17 of the revised manuscript. As can be seen, FlatQuant can achieve flat distribution for weights and activations across different linear layers.
>
>
> ---
> **Q8**. *I am curious about the speedup ratio of the method on Hopper architecture GPUs (e.g., H100, H800).*
>
> **A8.** We thank the reviewer for the question. Unfortunately, we currently lack access to Hopper architecture GPUs (e.g., H100, H800) and are unable to provide speedup results for these devices at this time. However, we are committed to extending our evaluations to more GPU architectures, including Hopper, as soon as these resources become available.
>
>
> ---
> **Q9**. *Could the authors explain why the per-channel scaling is employed? A proper $\mathbf P'$ can contain the ability to balance outliers between the weights and activations, for example, $\mathbf P'=\mathbf P\text{dig}(\mathbf c)$.*
>
> **A9**. As stated in Section 3.1, FlatQuant directly learns the decomposed transformations. However, we observe that such decomposition can result in an incomplete diagonal structure in the transformation matrix, which may affect its ability to balance outliers between weights and activations. To address this issue, we explicitly introduce a learnable scaling vector before the decomposed transformation to encourage the balance of quantization difficulty between weights and activations.

---

> > ### Comment · Reviewer_tWUe · 2024-11-23
> >
> > Thanks for the authors' reply, which addresses most of my concerns. I retain my initial score.

---

> > > ### Comment · Reviewer_tWUe · 2024-11-27
> > >
> > > After concisely viewing the comments from other reviewers, e.g., X7ak and RQEg, I decided to lower the score (the same concerns as theirs).
> > >
> > > Note: I seemed to be impulsive at the beginning because of the author's performance description in the abstract.

---

> > > > ### Author Response · Authors · 2024-11-28
> > > > **Response to Reviewer tWUe (part 4)**
> > > >
> > > > **Q10**. *After concisely viewing the comments from other reviewers, e.g., X7ak and RQEg, I decided to lower the score (the same concerns as theirs).*
> > > >
> > > >
> > > > **A10**. In summary, FlatQuant has achieved the best performance and speed-up compared to other quantization approaches on low bit (e.g. W4A4) quantization. In terms of the overhead, FlatQuant is in fact comparably fast with QuaRot (without kernel fusion) and faster (with kernel fusion). The additional memory overhead for affine transformation matrices is also minor (e.g., around 0.1M for one Transformer layer in LLaMa2-7B).
> > > >
> > > >
> > > > Please refer to our detailed response to Reviewer X7ak and RQEg for further clarification. Finally, please don't hesitate to let us know if there is any remaining concern, and if there is anything else we can do to help improve the score! Thank you!

---

> ### Author Response · Authors · 2024-12-02
>
> Dear Reviewer,
>
> Thank you for the time and attention you've dedicated to reviewing our paper. As the discussion stage is coming to an end, please let us know whether we have addressed your concerns or if any further discussions are needed.
>
> We look forward to your feedback and thank you once again for your valuable contribution to our work.
>
> Best, Authors

---

> ### Author Response · Authors · 2024-12-04
> **Still waiting for your response**
>
> Dear Reviewer tWUe,
>
> We are still waiting for your response to our feedback and open to any of your concerns.
>
> Thanks! Authors of FlatQuant.

---

### Official Review · Reviewer_X7ak · 2024-10-23

**Soundness:** 2
**Presentation:** 2
**Contribution:** 3
**Rating:** 5
**Confidence:** 4

**Summary:**

This paper builds upon AffineQuant by leveraging Kronecker decomposition to design efficient kernels that integrate affine transformations and quantization. This approach helps mitigate the doubled computation and memory access overhead caused by the pre-quantization transformation in AffineQuant. Experimental results demonstrate that FlatQuant achieves superior accuracy compared to existing methods under the W4A4 configuration.

**Strengths:**

* The paper employs Kronecker decomposition to design efficient kernels that integrate affine transformations and quantization, reducing the doubled computational and memory overhead caused by the pre-quantization transformation in AffineQuant.

* The paper further enhances accuracy by introducing activation clipping through learning, which yields notable improvements.

**Weaknesses:**

* Even with Kronecker decomposition, FlatQuant still requires a pre-quantization step before each linear input, leading to O(n) memory accesses and O(nlogn) computational overhead, where n is the model's feature dimension. For a single block of LLaMA, this transformation needs to be applied more than six times, increasing the model’s complexity. In contrast, SpinQuant requires only one pre-quantization transformation, which may offer a more significant advantage.  I would be very grateful if more comparisons on the additional computational complexity and parameter quantity improvement brought by Spinquant and FlatQuant could be provided.

* AffineQuant ensures that affine transformations remain invertible by employing diagonal dominance, but it’s unclear how FlatQuant maintains invertibility.  This information seems to be missing from the current paper and would be important for understanding the method's robustness.

* It is also puzzling that the paper does not report zero-shot performance for weight-only quantization. As shown in other works, LWC tends to cause overfitting, and the perplexity metric may not reflect true performance.  I would be very grateful if more weight-only results could be provided.

**Questions:**

* OmniQuant only trains smooth parameters, while FlatQuant adds the inverse affine matrix, pre-quantization, SVD decomposition, and learnable activation clipping operations. Strangely, the paper reports that FlatQuant is 10 times faster than OmniQuant, which raises concerns. A more equitable and transparent comparison will be employed if a more detailed breakdown of the timing comparisons is provided, including information on the implementation details and the hardware used for both methods.

---

> ### Author Response · Authors · 2024-11-23
> **Response to Reviewer X7ak (part 1)**
>
> We thank the reviewer for the thorough review of our paper and for offering valuable feedback. In response, we have provided additional explanations and experimental results to address each of the points raised.
>
>
> ---
> **Q1**. *Even with Kronecker decomposition, FlatQuant still requires a pre-quantization step before each linear input, leading to O(n) memory accesses and O(nlogn) computational overhead, where n is the model's feature dimension. For a single block of LLaMA, this transformation needs to be applied more than six times, increasing the model's complexity. In contrast, SpinQuant requires only one pre-quantization transformation, which may offer a more significant advantage. I would be very grateful if more comparisons on the additional computational complexity and parameter quantity improvement brought by Spinquant and FlatQuant could be provided.*
>
>
> **A1**. We would like to highlight that FlatQuant can achieve notable accuracy improvement compared with SpinQuant as shown in Table 1 and Table 2, while binging minimal inference overhead (e.g., <0.01x speedup loss). In the following, we show the speed-up comparison between FlatQuant and SpinQuant.
>
> * **Speedup Comparison with SpinQuant**. We summarize the speedup of FlatQuant and SpinQuant after incorporating different online transformations in the following table. It can be seen that with the designed efficient kernel, FlatQuant can achieve better speedup than SpinQuant.
>
> |           | +$\mathbf P_h$ | +$\mathbf P_o$ | +$\mathbf P_d$ | +$\mathbf P_a$ | +$\mathbf P_{ug}$ | Speedup | Speedup Gap with INT4 |
> | :-------- | :--------------- | :--------------- | :--------------- | :--------------- | :------------------ | :------ | :-------------------- |
> | SpinQuant | 2.37x            | -                | 2.21x            | -                | -                   | 2.21x   | 0.16x                 |
> | FlatQuant | 2.36x            | 2.35x            | 2.31x            | 2.30x            | 2.30x               | 2.30x   | 0.07x                 |
>
> * **Memory Consumption Comparison**. Both FlatQuant and SpinQuant incur negligible memory overhead. We compute the parameter count of each online transformation below: (1) $\mathbf{P}_a: 2 (\sqrt{h_d})^2$; (2) $\mathbf{P}_o: a^2$; (3) $\mathbf{P}_h: (h_d / a)^2$; (4) $\mathbf{P}_u\ _g: 2 (\sqrt{h_d})^2$; (5) $\mathbf{P}_d: 2 (\sqrt{h_i})^2$, where $h_d$ denotes the hidden dimension, $h_i$ denotes the intermediate dimension, $a$ denotes the number of attention heads. The additional memory consumption brought by FlatQuant in one Transformer layer during inference is $2(4h_d+2h_i+a^2+(h_d/a)^2)$ bytes, which only consumes about 0.11MB extra memory space for LLaMA-2-7B. The memory overhead for SpinQuant is $2((h_d / a)^2+2h_i)$ bytes, which only amounts to 0.07MB for LLaMA-2-7B.
>
>
> ---
> **Q2**. *AffineQuant ensures that affine transformations remain invertible by employing diagonal dominance, but it's unclear how FlatQuant maintains invertibility. This information seems to be missing from the current paper and would be important for understanding the method's robustness.*
>
>
> **A2**. We thank the reviewer for raising this important question. This is already discussed in Appendix B.1, where FlatQuant ensures invertibility by employing a singular value decomposition (SVD) re-parameterization trick during training. Specifically,
>
> * **The SVD Re-parameterization**. For a learnable affine transformation matrix $\mathbf P$, we decompose it as $\mathbf P=\mathbf U\Sigma\mathbf V^\top$, where  $\mathbf U,\mathbf V$ are orthogonal matrices, and $\Sigma$ is a diagonal matrix. We ensure the invertibility of $\mathbf P$ by forcing the elements of $\Sigma$ larger than zero. The optimization of orthogonal matrices can be readily accomplished with Cayley parameterization. Therefore, $\mathbf P^{-1}$ can be easily computed with $\mathbf V\Sigma^{-1}\mathbf U^\top$.
>
> * **Comparison with Direct Inversion**. In Appendix B.1, we also compare this SVD training strategy with the naive strategy of directly inverting the matrix $\mathbf P$, which cannot ensure the invertibility of $\mathbf P$ theoretically. Although we empirically find that this naive strategy can still produce invertible matrices in the training process, it requires FP32 training to stabilize the optimization process. After thorough comparisons between these two strategies, we finally adopt the SVD training which enjoys both training stability and efficiency.

---

> ### Author Response · Authors · 2024-11-23
> **Response to Reviewer X7ak (part 2)**
>
> **Q3**. *It is also puzzling that the paper does not report zero-shot performance for weight-only quantization. As shown in other works, LWC tends to cause overfitting, and the perplexity metric may not reflect true performance. I would be very grateful if more weight-only results could be provided.*
>
>
> **A3**. We thank the reviewer for the constructive suggestion. FlatQuant does not introduce significant overfitting on the calibration dataset and achieves consistent results across various downstream tasks.
>
> * Below, we provide the detailed zero-shot performance of FlatQuant for weight-only per-channel quantization on LLaMA3-8B as a complement to Table 5.
>
> | **LLaMA3-8B** | **WikiText-2** | **C4** | **ARC-C** | **ARC-E** | **HellaSwag** | **LAMBADA** | **PIQA** | **Winogrande** | **Avg.** |
> | :------------ | :------------- | :----- | :-------- | :-------- | :------------ | :---------- | :------- | :------------- | :------- |
> | FP16          | 6.14           | 9.45   | 53.50     | 77.57     | 79.12         | 75.51       | 80.74    | 72.93          | 73.23    |
> | W4A16         | 6.54           | 10.17  | 51.19     | 77.31     | 77.74         | 75.10       | 79.43    | 73.32          | 72.35    |
> | W3A16         | 7.78           | 12.64  | 46.76     | 73.32     | 74.09         | 69.61       | 79.33    | 71.51          | 69.10    |
>
>
> ---
>
> **Q4**. *OmniQuant only trains smooth parameters, while FlatQuant adds the inverse affine matrix, pre-quantization, SVD decomposition, and learnable activation clipping operations. Strangely, the paper reports that FlatQuant is 10 times faster than OmniQuant, which raises concerns. A more equitable and transparent comparison will be employed if a more detailed breakdown of the timing comparisons is provided, including information on the implementation details and the hardware used for both methods.*
>
> **A4**. We have reviewed our paper and confirmed that we did not claim FlatQuant is 10 times faster than OmniQuant. In fact, we believe the calibration time required by both methods is comparable, typically one hour on a single GPU for a 7B model (Line 339). Below, we explain in detail why FlatQuant does not introduce more calibration overhead compared to OmniQuant.
>
> * **Calibration Time Analysis**. FlatQuant leverages the observation that Kronecker decomposed transformation matrices are similarly effective in flattening activations and weights. As shown in Figure 5, this approach achieves comparable perplexity results. Consequently, FlatQuant directly learns the decomposed transformations, making it computationally efficient, with minimal extra overhead during the calibration stage.  Additionally, our implementation of SVD uses two orthogonal matrices and a vector to construct a transformation matrix. This implementation incurs an additional 9% calibration time compared to directly optimizing one matrix.

---

> > ### Comment · Reviewer_X7ak · 2024-11-26
> >
> > Thanks to the authors for their response.
> > - It appears that the performance of W4-only has not met my expectations, as it fails to maintain over 99% of the floating-point performance.
> > - I also maintain my position that introducing additional parameters for each linear layer is cumbersome. The reported speedup relative to SpinQuant primarily stems from the faster performance of Tensor Cores compared to CUDA Cores, rather than any advantage in parameter count or computational efficiency.
> >
> > Therefore, I consider this work an improved version of AffineQuant in terms of accuracy and speed. I will retain my original score.

---

> > > ### Author Response · Authors · 2024-11-28
> > > **Response to Reviewer X7ak (part 3)**
> > >
> > > **Q5**. *It appears that the performance of W4-only has not met my expectations, as it fails to maintain over 99% of the floating-point performance.*
> > >
> > > **A5**. First of all, we need to highlight that **FlatQuant is primarily designed for weight-activation quantization** (W4A4, in particular), as the learned affine transformations are particularly helpful to remove **outliers over activations** (as also visualized in Figure 1).
> > >
> > > Back to W4-only quantization, it is important to note that many W4-only quantization methods within 99% performance are primarily based on **per-group quantization** [1, 2, 3]. However, our setting focuses on **per-channel quantization**, which tends to suffer more from performance degradation. Below, we also provide results of per-channel quantization of GPTQ. It can be found that:
> > >
> > > * GPTQ with per-channel quantization also fails to achieve 99% performance.
> > >
> > > * Unlike FlatQuant-RTN, **FlatQuant-GPTQ and FlatQuant-RTN-g128 successfully maintain 99% of the floating-point performance**.
> > >
> > > | **LLaMA3-8B W4A16** | **WikiText-2** | **C4** | **ARC-C** | **ARC-E** | **HellaSwag** | **LAMBADA** | **PIQA** | **Winogrande** | **Avg.** |
> > > | :------------------ | :------------- | :----- | :-------- | :-------- | :------------ | :---------- | :------- | :------------- | :------- |
> > > | FP16                | 6.14           | 9.45   | 53.50     | 77.57     | 79.12         | 75.51       | 80.74    | 72.93          | 73.23    |
> > > | GPTQ                | 7.00           | 11.80  | 51.37     | 75.84     | 76.41         | 72.75       | 79.76    | 73.09          | 71.54    |
> > > | FlatQuant-RTN       | 6.54           | 10.17  | 51.19     | 77.31     | 77.74         | 75.10       | 79.43    | 73.32          | 72.35    |
> > > | FlatQuant-RTN-g128  | 6.53           | 10.21  | 51.96     | 77.82     | 77.90         | 74.31       | 80.52    | 73.09          | 72.60    |
> > > | FlatQuant-GPTQ      | 6.48           | 10.28  | 51.71     | 77.86     | 77.87         | 74.71       | 80.09    | 73.72          | 72.66    |
> > >
> > >
> > > [1] Frantar, Elias, et al. "OPTQ: Accurate quantization for generative pre-trained transformers." *The Eleventh International Conference on Learning Representations*. 2022.
> > >
> > > [2] Lin, Ji, et al. "AWQ: Activation-aware Weight Quantization for On-Device LLM Compression and Acceleration." *Proceedings of Machine Learning and Systems* 6 (2024): 87-100.
> > >
> > > [3] Huang, Wei, et al. "How good are low-bit quantized llama3 models? an empirical study." *arXiv e-prints* (2024): arXiv-2404.
> > >
> > >
> > > ---
> > >
> > > **Q6**. *I also maintain my position that introducing additional parameters for each linear layer is cumbersome.*
> > >
> > >
> > > **A6**. While FlatQuant introduces additional parameters for transformation, they are in fact quite **lightweight**, with only 2.61% overhead (FLOPs) and comparable speedup with QuaRot/SpinQuant even without kernel fusion (as detailed in Figure 6 of the revised manuscript).
> > >
> > >
> > > It's more like **a "free lunch" in FlatQuant**: you get comparable speed (without kernel fusion) and even faster speed (with kernel fusion), while the accuracy of quantized LLMs is improved significantly.
> > >
> > >
> > > ---
> > >
> > > **Q7**. *The reported speedup relative to SpinQuant primarily stems from the faster performance of Tensor Cores compared to CUDA Cores, rather than any advantage in parameter count or computational efficiency.*
> > >
> > > **A7**. To provide a fair comparison of the additional overhead from online transformations, we have tested both methods under the same quantization setting (i.e., per-channel symmetric quantization for weights and per-token symmetric quantization for activations) and on the same hardware, as shown in the table below. It is worth noting that SpinQuant actually uses per-group symmetric quantization for the weights of the output projection (o_proj) and asymmetric quantization for activations. This introduces additional overhead, resulting in a worse speedup than what is shown in the table. Moreover, compared to SpinQuant, FlatQuant offers a key advantage that should not be overlooked: **the significant improvement in accuracy**.
> > >
> > > |                             | +$\mathbf P_h$ | +$\mathbf P_o$ | +$\mathbf P_d$ | +$\mathbf P_a$ | +$\mathbf P_{ug}$ | Speedup | Speedup Gap with INT4 |
> > > | :-------------------------- | :------------- | :------------- | :------------- | :------------- | :---------------- | :------ | :-------------------- |
> > > | SpinQuant                   | 2.37x          | -              | 2.21x          | -              | -                 | 2.21x   | 0.16x                 |
> > > | FlatQuant w/o Kernel Fusion | 2.36x          | 2.29x          | 2.19x          | 2.15x          | 2.11x             | 2.11x   | 0.26x                 |
> > > | FlatQuant w/ Kernel Fusion  | 2.36x          | 2.35x          | 2.31x          | 2.30x          | 2.30x             | 2.30x   | 0.07x                 |

---

> ### Author Response · Authors · 2024-12-03
>
> Dear Reviewer,
>
> Thank you for the time and attention you've dedicated to reviewing our paper. As the discussion stage is coming to an end, please let us know whether we have addressed your concerns or if any further discussions are needed.
>
> We look forward to your feedback and thank you once again for your valuable contribution to our work.
>
> Best, Authors

---

> ### Author Response · Authors · 2024-12-04
> **Still waiting for your response**
>
> Dear Reviewer X7ak,
>
> We are still awaiting your response to our feedback. Specifically, we have demonstrated above that for W4-only quantization, both FlatQuant-GPTQ and FlatQuant-RTN-g128 can successfully retain 99% of the fp16 performance; FlatQuant also achieves considerable speed-up compared with SpinQuant even without the kernel fusion. We are eager to address any concerns you may have.
>
> Best regards,
> The Authors of FlatQuant.

---

### Official Review · Reviewer_RQEg · 2024-10-30

**Soundness:** 2
**Presentation:** 3
**Contribution:** 2
**Rating:** 3
**Confidence:** 5

**Summary:**

This paper proposes FLATQUANT, a fast learnable affine transformation used to enhance the smoothing of weights and activation values, while the authors propose the use of Kronecker decomposition and operator fusion to reduce the runtime burden. Extensive experiments demonstrate the effectiveness of FLATQUANT in the W4A4 case.

**Strengths:**

1. The paper was well written and clearly presented;
2. The authors show the effectiveness of the algorithm through extensive experiments;
3. The authors reduce the runtime burden by means of Kronecker decomposition and operator fusion, which is meaningful for many scenarios.

**Weaknesses:**

Although the author's method achieved significant gains, there were a few points that could not be ignored that contributed to my initial score:
1. The flatness and quantization error expressed in the authors' paper do not strictly correspond. A simple example [3,4,5,6] has significantly higher quantization error at 2-bit than [0,0,0,sqrt(86)]; in fact, the authors only optimize the loss in Eq. (4) in their paper, and there is no direct correlation with flatness;
2. In the paper, the authors mainly compare the method with QuaRot and SpinQuant, and from my point of view, the way that authors compare is unfair, because the computational overhead of FLATQUANT is totally different from QuaRot and SpinQuant. Taking QuaRot as an example, only the rotation matrix in Query and Key and the rotation matrix after SiLU need to participate in the online computation in QuaRot, while all other matrices can be merged into the weights, while FLATQUANT has a large number of matrix multiplications that need to participate in the matrix computation of the floating-point, which introduces quite a lot of computational burdens compared to QuaRot, so I think the author's comparison is totally unfair. Meanwhile, the authors did not clarify this part of the paper;
3. While I strongly agree that the author's use of Kronecker decomposition and operator fusion reduces the runtime burden, which is also valid for QuaRot, the author's Speed Up comparison of Online Transformations in Figure.6 leads readers to mistakenly believe that FLATQUANT's accuracy-performance of the In fact, on the one hand, QuaRot's additional floating-point computation is less than FLATQUANT's, and on the other hand, the Kronecker decomposition and operator fusion reduces Hadamard's runtime burden under GEMM computation (instead of Walsh-Hadamard Transformation), so I think the comparison in Figure.6 is biased and misleading.

In summary, while I agree with the Kronecker decomposition and operator fusion approach used by the authors to minimize the burden on the online runtime, given the problems with the authors' motivation and experimental comparisons, I give my scores.

**Questions:**

See weekness.

---

> ### Author Response · Authors · 2024-11-23
> **Response to Reviewer RQEg (part 1)**
>
> We note that the reviewer's concerns mainly focus on two aspects: (1) the relationship between flatness and quantization error, and (2) the computational overhead comparison with QuaRot and SpinQuant.
>
>
> To address these concerns, we would like to clarify some key observations to resolve potential misunderstandings, and then respond to each question in detail.
>
> 1. Although FlatQuant does not explicitly optimize for flatness, our results in Figure 1 and 2 demonstrate that **the learned transformations naturally flatten weights and activations**.
>
> 2. We further quantify and plot the flatness changes during the learning process, as shown in the Appendix C.1 of the revised manuscript (Line 906~936). The flatness curves consistently and smoothly decrease as optimization progresses, indicating that **FlatQuant promotes flatness**, which is the underlying reason for FlatQuant's flat quantization error surface.
>
> 3. FlatQuant is designed **lightweight**, with computational overhead comparable to QuaRot and SpinQuant, both with and without kernel fusion.
>
> 4. We appreciate the reviewer highlighting the Kronecker decomposition and **operator fusion** in our method and their potential generalizability to methods like QuaRot and SpinQuant. This is one of our key contributions, and we are grateful for the recognition.
>
> 5. FlatQuant achieves a significant **accuracy boost** with only marginal inference overhead.
>
>
> ---
> **Q1**. *The flatness and quantization error expressed in the authors' paper do not strictly correspond. A simple example [3,4,5,6] has significantly higher quantization error at 2-bit than [0,0,0,sqrt(86)];*
>
> **A1**. We appreciate the reviewer's observation. While the example provided is mathematically valid, it represents an edge case that does not generalize to practical LLM quantization scenarios. For LLMs, millions of parameters in each layer rarely align perfectly, and a lower step size resulting from flatness offers significant benefits.
>
> **General Relationship Between Flatness and Quantization Error.** A flatter distribution reduces the quantization range, which in turn minimizes the step size and directly decreases quantization error. In LLMs, the majority of weights and activations do not align perfectly with quantization grid points. Consequently, flatter distributions are essential for reducing expected rounding errors and improving overall quantization efficiency.
>
>
> ---
> **Q2**. *... in fact, the authors only optimize the loss in Eq. (4) in their paper, and there is no direct correlation with flatness;*
>
> **A2.** We thank the reviewer for raising this point. While we do not explicitly optimize flatness, our results indicate the optimization of the simple layer-wise quantization loss defined in Equation 4 is enough to encourage flatness in weights and activations, which is the underlying reason for FlatQuant's flat quantization error surface.
>
> * **Empirical Evidence.** Figure 1 and 2 demonstrate that the learned transformations naturally flatten weights and activations. This suggests that minimizing the layer-wise MSE loss promotes flatness.
>
> * **FlatQuant Leads to Flatness.** To further address this concern, we have quantified flatness and plotted its evolution during optimization in Figure 8 with details in Appendix C.1 (Line 906~936) of the revised manuscript. The results show that the flatness of weights and activations (measured as their deviation from a perfectly flat distribution) improves smoothly as optimization progresses. This provides further evidence that the affine transformations of FlatQuant actually learn to flatten the distribution of weights and activations to lower the quantization error, even though we do not include flatness in the optimization target.

---

> ### Author Response · Authors · 2024-11-23
> **Response to Reviewer RQEg (part 2)**
>
> **Q3**.*In the paper, the authors mainly compare the method with QuaRot and SpinQuant, and from my point of view, the way that authors compare is unfair, because the computational overhead of FLATQUANT is totally different from QuaRot and SpinQuant. Taking QuaRot as an example, only the rotation matrix in Query and Key and the rotation matrix after SiLU need to participate in the online computation in QuaRot, while all other matrices can be merged into the weights, while FLATQUANT has a large number of matrix multiplications that need to participate in the matrix computation of the floating-point, which introduces quite a lot of computational burdens compared to QuaRot, so I think the author's comparison is totally unfair. Meanwhile, the authors did not clarify this part of the paper;*
>
>
> **A3**. There can be some misunderstanding of our approach. A key idea in FlatQuant is to ensure the online transformations are lightweight. Despite more transformations are introduced in FlatQuant, its computational overhead is still comparable to that of QuaRot or SpinQuant. While QuaRot merges most matrices into weights, FlatQuant's optimizations, such as Kronecker decomposition (see Section 3.1), ensure that the runtime burden is effectively mitigated.
>
> * **Computational FLOPs.** FlatQuant's online transformations contribute only **2.61% of the total FLOPs** of FP16 LLaMA-2-7B, ensuring a low computational overhead, as detailed in Appendix C.5. While FlatQuant includes two more transformations than QuaRot (i.e., $\mathbf{P_a}, \mathbf{P_{ug}}$), they are lightweight, contributing only **23.10% of the total extra FLOPs** for online transformations. This results in a negligible latency increase (0.08× without kernel fusion, 0.01× with kernel fusion), please refer to the table in A4 below for details.
>
> * **End-to-End Latency.** In practice, the actual end-to-end speedup difference between FlatQuant and naive INT4 quantization is minimal (<0.10×) during both prefill and decoding stages, indicating that the introduced computational overhead is negligible.
>
> * **Fairness of Comparison.** We compare the speed-up between FlatQuant (both with and without kernel fusion) and QuaRot in the tables of A4 below. The empirical results further validate that FlatQuant's computational overhead is on par with QuaRot without kernel fusion, and surpass QuaRot after kernel fusion.

---

> ### Author Response · Authors · 2024-11-23
> **Response to Reviewer RQEg (part 3)**
>
> **Q4**. *While I strongly agree that the author's use of Kronecker decomposition and operator fusion reduces the runtime burden, which is also valid for QuaRot, the author's Speed Up comparison of Online Transformations in Figure.6 leads readers to mistakenly believe that FLATQUANT's accuracy-performance of the In fact, on the one hand, QuaRot's additional floating-point computation is less than FLATQUANT's, and on the other hand, the Kronecker decomposition and operator fusion reduces Hadamard's runtime burden under GEMM computation (instead of Walsh-Hadamard Transformation), so I think the comparison in Figure.6 is biased and misleading.*
>
>
> **A4**. We thank the reviewer for the positive recognition of our use of Kronecker decomposition and operator fusion. Below, we provide additional analysis to clarify potential misunderstandings about Figure 6:
>
> * **Speedup Without Kernel Fusion.** We conduct a comparison of speedup without kernel fusion against QuaRot. The table below (also revised in Figure 6 of the revised manuscript) demonstrates that even without kernel fusion, FlatQuant achieves comparable speedup to QuaRot. This is primarily because the Kronecker decomposition and highly optimized GEMM kernel on GPUs.
>
> |           | +$\mathbf P_h$ | +$\mathbf P_o$ | +$\mathbf P_d$ | +$\mathbf P_a$ | +$\mathbf P_{ug}$ | Speedup | Speedup Gap with INT4 |
> | :-------- | :------------- | :------------- | :------------- | :------------- | :---------------- | :------ | :-------------------- |
> | QuaRot    | 2.37x          | 2.27x          | 2.11x          | -              | -                 | 2.11x   | 0.26x                 |
> | FlatQuant | 2.36x          | 2.29x          | 2.19x          | 2.15x          | 2.11x             | 2.11x   | 0.26x                 |
>
> * **Contribution of Kernel Fusion.** Incorporating kernel fusion (detailed in Section 3.3) further reduces FlatQuant's inference overhead, achieving a speedup gap of only **0.07×** compared to naive INT4 quantization. We summarize the results in Figure 6 in the following table.
>
> |           | +$\mathbf P_h$ | +$\mathbf P_o$ | +$\mathbf P_d$ | +$\mathbf P_a$ | +$\mathbf P_{ug}$ | Speedup | Speedup Gap with INT4 |
> | :-------- | :------------- | :------------- | :------------- | :------------- | :---------------- | :------ | :-------------------- |
> | QuaRot    | 2.37x          | 2.27x          | 2.11x          | -              | -                 | 2.11x   | 0.26x                 |
> | FlatQuant | 2.36x          | 2.35x          | 2.31x          | 2.30x          | 2.30x             | 2.30x   | 0.07x                 |
>
> * **WHT Has Similar Latency with GEMM.** Despite WHT having fewer FLOPs than the vanilla matrix multiplication with GEMM, it can only be applied to cases where the size is an integer power of 2. In practice, we find that the `fast_hadamard_transformation` library used in QuaRot offers a similar speed-up compared to GEMM (e.g., both about 0.02 ms when the input shape is $2048\times 4096$). This is because the GEMM kernel is highly optimized on the low-level with parallelism on Tensor Cores.
>
> * **Other Limitations of Hadamard Matrices.** Hadamard transformations are not well-defined for all sizes. Manually defining Hadamard matrices for special cases, such as the intermediate size of Qwen-2.5-7B, can be challenging and may hinder their applicability. Additionally, Hadamard matrices can limit the expressiveness of the transformation and thus affect the accuracy of quantization.

---

> ### Comment · Reviewer_RQEg · 2024-11-26
>
> I also agree with official comment by Reviewer X7ak. FlatQuant should be compared with AffineQuant, DuQuant[1] (NeurIPS2024, so it is not necessary). Although **I strongly agree with the effectiveness of the methodology, I still think the comparison of the author's experiment problematic.**
>
> I understand rotational invariance very well, and the enhancement of the author's method compared to rotational invariance is mainly brought about by the fact that the author did not follow rotational invariance.
>
> Therefore, I want to give 4 score, however there is no choice. So I will retain my original score.
>
> By the way, the author is provided triton kernel, not cuda. It seems both reviewer X7ak and tWUe make mismake.
>
>
> [1] Lin, H., Xu, H., Wu, Y., Cui, J., Zhang, Y., Mou, L., Song, L., Sun, Z. and Wei, Y., 2024. Rotation and Permutation for Advanced Outlier Management and Efficient Quantization of LLMs. arXiv preprint arXiv:2406.01721.

---

> > ### Author Response · Authors · 2024-11-28
> > **Response to Reviewer RQEg (part 4)**
> >
> > **Q5**. *FlatQuant should be compared with AffineQuant, DuQuant[1] (NeurIPS2024, so it is not necessary).*
> >
> > **A5.** In the following, we further compare FlatQuant against AffineQuant and DuQuant in terms of accuracy and speedup. We hope this clarifies any concern regarding the experimental comparison, as our results demonstrate **clear advantages in both effectiveness and efficiency**.
> >
> > * **Comparison with AffineQuant.** In Table 1, AffineQuant is already compared with FlatQuant in our manuscript, where FlatQuant outperforms AffineQuant by a large margin (e.g. 6.90 in PPL on LLaMA-2-7B and 6.33 on LLaMA-2-13B).
> >
> > * **Comparison with DuQuant.** The comparisons with DuQuant are listed below. DuQuant has the same number of online transformations (each consisting of two matrix multiplications and one channel permutation) as FlatQuant. As shown, FlatQuant significantly outperforms DuQuant across multiple benchmarks. Notably, FlatQuant achieves comparable speed without kernel fusion and even faster speed with kernel fusion.
> >
> > | **LLaMA3-8B** | **WikiText-2** | **C4** | **ARC-C** | **ARC-E** | **HellaSwag** | **LAMBADA** | **PIQA** | **Winogrande** | **Avg.** |
> > | :------------ | :------------- | :----- | :-------- | :-------- | :------------ | :---------- | :------- | :------------- | :------- |
> > | FP16          | 6.14           | 9.45   | 53.50     | 77.57     | 79.12         | 75.51       | 80.74    | 72.93          | 73.23    |
> > | DuQuant       | 8.13           | 12.91  | 44.80     | 71.30     | 73.00         | 68.04       | 75.73    | 69.46          | 67.05    |
> > | FlatQuant     | 6.98           | 11.13  | 50.00     | 75.80     | 76.80         | 72.91       | 79.16    | 72.69          | 71.23    |
> >
> >
> > | **Batch Size**              | **1** | **4** | **16** |
> > | :-------------------------- | :---- | :---- | :----- |
> > | DuQuant                     | 1.95x | 2.03x | 2.08x  |
> > | FlatQuant w/o Kernel Fusion | 1.94x | 2.02x | 2.10x  |
> > | FlatQuant w/ Kernel Fusion  | 2.12x | 2.21x | 2.27x  |
> >
> >
> > ---
> >
> > **Q6**. *Although I strongly agree with the effectiveness of the methodology, I still think the comparison of the author's experiment problematic.*
> >
> > **A6**. As detailed in **A3 and A4**, FlatQuant is quite lightweight, with only 2.61% overhead (FLOPs) and comparable speedup with **QuaRot**/**SpinQuant** even without kernel fusion. In **A5**, we further demonstrate that FlatQuant is comparable with **DuQuant** in terms of efficiency.
> >
> >
> > Based on these results, we believe **our experiments are based on fair and rigorous comparisons**. Please don't hesitate to let us know if there are any remaining concerns.
> >
> >
> > ---
> >
> > **Q7**. *I understand rotational invariance very well, and the enhancement of the author's method compared to rotational invariance is mainly brought about by the fact that the author did not follow rotational invariance.*
> >
> > **A7**. While we do not fully understand your description here, we still need to point out that **computational invariance** is the key to enhancing LLM quantization, and rotational invariance is one way to achieve that. Another way to computational invariance is affine transformation with invertible matrices, which has been shown to significantly improve the accuracy of quantized LLMs with little inference overhead, as verified in our work.

---

> > ### Author Response · Authors · 2024-11-28
> > **Response to Reviewer RQEg (part 5)**
> >
> > **A8**.**Finally, we have carefully prepared responses to your queries in the review. Please let us know if they address your concerns, and what else can be done that can help raise the score. Thank you!**

---

> ### Author Response · Authors · 2024-12-02
>
> Dear Reviewer,
>
> Thank you for the time and attention you've dedicated to reviewing our paper. As the discussion stage is coming to an end, please let us know whether we have addressed your concerns or if any further discussions are needed.
>
> We look forward to your feedback and thank you once again for your valuable contribution to our work.
>
> Best, Authors

---

> ### Author Response · Authors · 2024-12-04
> **Still waiting for your response**
>
> Dear Reviewer RQEg,
>
> We are still waiting for your response to our feedback and open to any of your concerns.
>
> Thanks!
> Authors of FlatQuant.

---

### Official Review · Reviewer_YdzG · 2024-11-01

**Soundness:** 3
**Presentation:** 3
**Contribution:** 2
**Rating:** 5
**Confidence:** 4

**Summary:**

This paper propose FLATQUANT to enhance flatness of weight and activation. It consist of:

1) Learnable affine transformation;
2) Kronecker Decomposition to decompose the large transformation into 2 small matrix;
3) Per-Channel Scaling
4) Learnable Clipping Thresholds.

This paper is easy to follow. Experiments show relatively good performance compared to current SOTA like SpinQuant. Ablation study part explains  the effect of each technique.

**Strengths:**

1. This paper propose Kronecker decomposition to decompose the large matrix P into 2 small matrix. The varying size of decomposition  have limited impact on the perplexity of generated text, and the speedup peaks when P1 and P2 are of equal size.
2. This paper propose learnable clipping threshold, where current works usually adopts grid search.
3. The figures in this paper helps easy understanding of the proposed FLATQUANT.

**Weaknesses:**

1. In perplexity, learnable transformation and learnable scaling of FLATQUANT are borrowed from existing works.
2. Learnable clipping threshold does not compare with existing CNN works like NWQ[1], also does not show detailed analysis on learnable clipping threshold  why not causing important outliers clipped, as Llm.int8() [2].
3. In speedup, Kronecker Decomposition still involves online transformation and can not achieve pure int computation.

[1] Leveraging Inter-Layer Dependency for Post -Training Quantization . Changbao Wang

[2] LLM.int8(): 8-bit Matrix Multiplication for Transformers at Scale. Tim Dettmers

**Questions:**

1. What's the perplexity improvement of pure learnable clipping threshold? In table3, in perplexity, whether the difference of SpinQuant and FLATQUANT is the learnable clipping threshold? Is the perplexity improvement mainly from pure learnable clipping threshold?
2. In Figure5, why the speedup curve is not almost-symmetricly distributed? The speedup of 2048 is substantially slower  than  2's.

---

> ### Author Response · Authors · 2024-11-23
> **Response to Reviewer YdzG (part 1)**
>
> We thank the reviewer for the valuable feedback. Below, we address the raised concerns in detail.
>
>
> ---
> **Q1**. *In perplexity, learnable transformation and learnable scaling of FLATQUANT are borrowed from existing works.*
>
> **A1**. We elaborate on the key differences between FlatQuant and previous works below:
>
> * **Kronecker Decomposed Matrices vs. Diagonally Dominant Matrices.** While AffineQuant optimizes strictly diagonally dominant matrices, FlatQuant uses unconstrained affine transformations to enhance flatness, minimizing quantization loss. With Kronecker decomposition, FlatQuant reduces computational overhead, enabling transformations in all linear layers to minimize the quantization loss, unlike AffineQuant's selective application due to higher costs.
>
> * **Learnable Scaling After Transformations.** A key difference of  learnable scaling in FlatQuant is that it is applied after the pre-quantization transformations. This can help avoid damaging critical outliers during activation clipping (e.g., LLM.int8()) and further improves accuracy. The detailed discussions on the order of learnable scaling and pre-quantization transformation are provided in Table 16 in Appendix C.3, and re-discussed in the response below to Q3.
>
>
> ---
> **Q2**. *Learnable clipping threshold does not compare with existing CNN works like NWQ.*
>
>
> **A2**. We thank the reviewer for the suggestion regarding related works. While our focus is primarily on LLM quantization, we would still clarify the differences from NWQ below:
>
> * **Different Research Focuses.** NWQ primarily addresses challenges related to gradient optimization in QAT, while FlatQuant focuses on PTQ by introducing fast and learnable affine transformations and learnable clipping thresholds to handle outliers.
>
> * **Orthogonality of These Two Methods.** While NWQ could potentially complement FlatQuant in a training setting, it does not address the unique challenges of PTQ for LLMs, such as handling activation and weight flatness. We believe the scope of NWQ is largely orthogonal to FlatQuant.
>
>
> ---
> **Q3.** *... also does not show detailed analysis on learnable clipping threshold why not causing important outliers clipped, as Llm.int8().*
>
> **A3**. We thank the reviewer for the inspiring question. FlatQuant preserves important outliers by applying the learnable clipping threshold (LCT) after the pre-quantization transformation. A detailed comparison of different clipping strategies is provided in Table 16 and Appendix C.3:
>
> * **"LCT before Transformation"** consistently performs worse, as it directly clips important outliers in LLMs.
>
> * **"LCT after Transformation"** performs significantly better because our fast and learnable affine transformations redistribute outliers across channels, making them less sensitive to clipping. Even after clipping, the inverse transformation can effectively recover the original outliers from the clipped activations.
>
> | **LLaMA3-8B**                | **WikiText-2** | **C4** | **ARC-C** | **ARC-E** | **HellaSwag** | **LAMBADA** | **PIQA** | **Winogrande** | **Avg.** |
> | :--------------------------- | :------------- | :----- | :-------- | :-------- | :------------ | :---------- | :------- | :------------- | :------- |
> | FP16                         | 6.14           | 9.45   | 53.50     | 77.57     | 79.12         | 75.51       | 80.74    | 72.93          | 73.23    |
> | w/o LCT                      | 7.95           | 12.74  | 44.20     | 71.89     | 74.21         | 68.72       | 77.15    | 66.30          | 67.08    |
> | LCT before Transformation    | 7.37           | 11.86  | 48.72     | 76.18     | 75.11         | 66.65       | 77.91    | 67.17          | 68.62    |
> | QuaRot-style Fixed Threshold | 7.25           | 11.62  | 48.21     | 75.29     | 75.66         | 71.32       | 78.73    | 70.01          | 69.87    |
> | LCT after Transformation     | 6.98           | 11.13  | 50.00     | 75.80     | 76.80         | 72.91       | 79.16    | 72.69          | 71.23    |
>
>
> ---
> **Q4**. *In speedup, Kronecker Decomposition still involves online transformation and can not achieve pure int computation.*
>
> **A4.** We thank the reviewer for the insightful comment. Although Kronecker decomposition involves online transformations, the computational overhead is minimal:
>
> * **Minimal Overhead.** Online transformations account for only 2.61% of the FP16 LLaMA-2-7B model's FLOPs, leading to a speedup gap of less than 0.10x compared to naive INT4 quantization (see Appendix C.5 Line 1075~1112 and Section 4.3).
>
> * **Accuracy Improvement.** FlatQuant achieves less than 1% accuracy drop on LLaMA-3-70B with W4A4KV4 quantization, making the computational cost negligible given the significant accuracy gain.

---

> > ### Comment · Reviewer_YdzG · 2024-11-25
> >
> > Thanks for authors' reply. I made a mistake for NWQ's reference, which should be "Leveraging Inter-Layer Dependency for Post -Training Quantization . Changbao Wang". Please compare learnable clipping threshold with existing similar CNN works like NWQ.

---

> > > ### Author Response · Authors · 2024-12-01
> > > **Response to Reviewer YdzG (part 3)**
> > >
> > > **Q8**.  *I made a mistake for NWQ's reference, which should be "Leveraging Inter-Layer Dependency for Post-Training Quantization . Changbao Wang". Please compare learnable clipping threshold with existing similar CNN works like NWQ.*
> > >
> > > **A8**. Thank the reviewer for the reference work. While much of NWQ's approach (i.e., AR, ASoftmax, AMixup) is orthogonal to FlatQuant, we make a comparison between LWC (Learnable Weight Clipping) in FlatQuant and ASoftmax (Annealing Softmax) in NWQ. For fair comparisons, we substitute the LWC with ASoftmax, and leave the rest components of FlatQuant unchanged. For **ASoftmax**, we set $n=0,m=1$, and $\tau$​​ is decayed from 1 to 0.01 as described in the NWQ paper. We grid search the best learning rate 1e-3 for the ASoftmax parameters. Our results are summarized in the table below:
> > >
> > > -  **LWC can achieve better performance on downstream tasks with much fewer learnable parameters than ASoftmax**. ASoftmax achieves further improvement relative to the baseline (FlatQuant w/o LWC). However, it requires learnable parameters that are twice the size of the model weight parameters, which makes it hard to be trained in an end-to-end fashion as done in NWQ. Besides, Baseline+ASoftmax has lower training loss but worse downstream performance compared to the Baseline+LWC. We suspect that the massive learnable parameters in ASoftmax may cause overfitting issues in the LLM PTQ setting.
> > >
> > > | **LLaMA3-8B**                | **Final Training Loss** | **WikiText-2** | **C4** | **ARC-C** | **ARC-E** | **HellaSwag** | **LAMBADA** | **PIQA** | **Winogrande** | **Avg.** |
> > > | :--------------------------- | :---------------------- | :------------- | :----- | :-------- | :-------- | :------------ | :---------- | :------- | :------------- | :------- |
> > > | FP16                         | -                       | 6.14           | 9.45   | 53.50     | 77.57     | 79.12         | 75.51       | 80.74    | 72.93          | 73.23    |
> > > | Baseline (FlatQuant w/o LWC) | 0.28764960              | 7.30           | 11.64  | 48.46     | 74.87     | 76.08         | 71.20       | 79.05    | 70.40          | 70.01    |
> > > | +ASoftmax                    | 0.20078197              | 7.17           | 11.57  | 49.49     | 77.99     | 76.13         | 71.76       | 79.11    | 70.48          | 70.83    |
> > > | +ASoftmax+LWC                | 0.17657542              | 6.95           | 11.15  | 50.34     | 75.38     | 76.43         | 72.70       | 78.67    | 71.51          | 70.84    |
> > > | +LWC                         | 0.22373830              | 6.98           | 11.13  | 50.00     | 75.80     | 76.80         | 72.91       | 79.16    | 72.69          | 71.23    |

---

> ### Author Response · Authors · 2024-11-23
> **Response to Reviewer YdzG (part 2)**
>
> **Q5**. *What's the perplexity improvement of pure learnable clipping threshold?*
>
> **A5**. We thank the reviewer for the insightful question. We have added ablation results for "LCT," "LT" (learnable transformation), and their combination ("LT+LCT") in the revised manuscript. The results demonstrate that while "LT" outperforms pure "LCT" by a large margin, their combination, combined with our clipping-after-transformation strategy, achieves the best performance.
>
> | **LLaMA3-8B** | **WikiText-2** | **C4**  | **ARC-C** | **ARC-E** | **HellaSwag** | **LAMBADA** | **PIQA** | **Winogrande** | **Avg.** |
> | :------------ | :------------- | :------ | :-------- | :-------- | :------------ | :---------- | :------- | :------------- | :------- |
> | FP16          | 6.14           | 9.45    | 53.50     | 77.57     | 79.12         | 75.51       | 80.74    | 72.93          | 73.23    |
> | LT            | 8.50           | 13.51   | 44.97     | 71.38     | 73.17         | 67.05       | 76.88    | 67.48          | 66.82    |
> | LCT           | 1149.08        | 1490.08 | 22.95     | 29.29     | 27.35         | 0.60        | 52.99    | 50.83          | 30.67    |
> | LT+LCT        | 7.11           | 11.47   | 49.32     | 76.14     | 76.30         | 72.17       | 78.89    | 71.51          | 70.72    |
>
>
> ---
> **Q6**. *In table3, in perplexity, whether the difference of SpinQuant and FLATQUANT is the learnable clipping threshold? Is the perplexity improvement mainly from pure learnable clipping threshold?*
>
> **A6**. **Fair Comparison with SpinQuant (without Weight/Activation Clipping).** We thank the reviewer for the valuable question. In the following, we make a fair comparison between FlatQuant and SpinQuant without weight or activation clipping. We reproduce experiments with FlatQuant and SpinQuant, without weight or activation clipping, under the same quantization settings (asymmetric activation and symmetric weight quantization). The results are summarized in the table below. Notably, SpinQuant relies heavily on GPTQ for maintaining accuracy, while FlatQuant achieves better accuracy with the RTN weight quantizer. This further highlights the superiority of FlatQuant.
>
> | **LLaMA3-8B**  | **WikiText-2** | **C4**    | **ARC-C** | **ARC-E** | **HellaSwag** | **LAMBADA** | **PIQA** | **Winogrande** | **Avg.**  |
> | :------------- | :------------- | :-------- | :-------- | :-------- | :------------ | :---------- | :------- | :------------- | :-------- |
> | FP16           | 6.14           | 9.45      | 53.50     | 77.57     | 79.12         | 75.51       | 80.74    | 72.93          | 73.23     |
> | SpinQuant-RTN  | 41.15          | 63.89     | 24.15     | 38.09     | 45.12         | 20.80       | 59.47    | 55.01          | 40.44     |
> | SpinQuant-GPTQ | 7.95           | 13.44     | 44.80     | 73.36     | 73.79         | 66.99       | 76.01    | 68.11          | 67.18     |
> | FlatQuant-RTN  | **7.71**       | **12.23** | 45.05     | 71.63     | 74.96         | 69.14       | 78.73    | 69.06          | **68.09** |
>
> **Effectiveness of LCT**. Besides, while SpinQuant does not use activation clipping due to marginal improvement, we find that learnable activation clipping, when combined with our method, leads to noticeable improvements as already discussed above.
>
>
> ---
> **Q7**. *In Figure5, why the speedup curve is not almost-symmetricly distributed? The speedup of 2048 is substantially slower than 2's.*
>
> **A7**. The asymmetry in the speedup curve arises from irregular memory access patterns, particularly when $n_1$ exceeds 64. Below is a detailed explanation:
>
> * As discussed in Section 4.3, left-multiplying the activation tensor by $\mathbf P_1\in\mathbb R^{n_1\times n_1}$ results in irregular memory access, making it more computationally expensive than right-multiplying by $\mathbf P_2\in\mathbb R^{n_2\times n_2}$, even though both have identical FLOPs.
>
> * In Figure 5, $n_1$ ranges from 1 to 2048. When $n_1 = 2048$, the activation tensor is left-multiplied by a $2048 \times 2048$ matrix and right-multiplied by a $2 \times 2$ matrix. This is significantly slower than the reverse case ($n_1 = 2$), where left-multiplication involves a $2 \times 2$ matrix and right-multiplication involves a $2048 \times 2048$ matrix.

---

> ### Author Response · Authors · 2024-12-02
>
> Dear Reviewer,
>
> Thank you for the time and attention you've dedicated to reviewing our paper. As the discussion stage is coming to an end, please let us know whether we have addressed your concerns or if any further discussions are needed.
>
> We look forward to your feedback and thank you once again for your valuable contribution to our work.
>
> Best,
> Authors

---

> ### Author Response · Authors · 2024-12-04
> **Still waiting for your response**
>
> Dear Reviewer YdzG,
>
> We are still wating for your response to our feedback, particularly the comparisons with NWQ.
>
> Thanks! Authors of FlatQuant

---

### Official Review · Reviewer_m9T2 · 2024-11-03

**Soundness:** 3
**Presentation:** 3
**Contribution:** 3
**Rating:** 8
**Confidence:** 5

**Summary:**

This paper introduces FlatQuant, a post-training quantization method for LLMs that focuses on flattening the distributions of weights and activations to reduce the quantization error. The key innovation is using learnable affine transformations optimized per layer, with Kronecker decomposition to reduce overhead. The authors also develop an efficient kernel that fuses the transformations with quantization.

The main contributions are:
1. A theoretical framework highlighting the importance of flatness in LLM quantization
2. The FlatQuant method that achieves state-of-the-art W4A4 quantization results
3. An efficient implementation that provides up to 2.3x speedup for prefill and 1.7x for decoding, with minimal overhead from transformations (0.07x vs QuaRot's 0.26x)

The approach is validated across multiple model sizes and quantization settings with comprehensive experiments and ablations.

**Strengths:**

1. The observation that flatness is the main influence on quantization error is insightful and well-supported. The motivation to improve flatness through learnable transformations is theoretically sound.

2. The Kronecker decomposition of the transformation matrix effectively reduces both memory and computation costs while maintaining performance. The training methodology is well-designed and practical.

3. The CUDA kernel implementation effectively reduces inference cost through fusion of operations.

4. The evaluation demonstrates that FlatQuant achieves impressive results, particularly: Less than 1% accuracy drop for W4A4 quantization on LLaMA-3-70B; Consistent improvements across different model sizes and tasks; Better speedups compared to existing methods like QuaRot.

**Weaknesses:**

1. Figure 3 is confusing and needs better explanation. The caption should better explain the relationships between subfigures and clarify the flow of operations.

2. While the efficient implementation and speedup is impressive, theoretical complexity analysis is crucial for algorithm development. The evaluation lacks important theoretical analysis:
- No detailed computation complexity analysis
- Missing information about total parameter counts (including decomposed transform matrices)
- No comprehensive memory consumption analysis for inference

3. Limited model architecture coverage (only LLaMA family in this paper)

4. Citation gap for learnable clipping threshold work, such as "Duanmu H, Yuan Z, Li X, et al. SKVQ: Sliding-window Key and Value Cache Quantization for Large Language Models[J]. arXiv preprint arXiv:2405.06219, 2024."

**Questions:**

1. Regarding the Kronecker decomposition: What is the perplexity performance without decomposition (direct training)?

2. Model generalization:
- How does the method perform on other model architectures (Gemini, Qwen, etc.)?
- Are there any architectural limitations or requirements?
- Would different model architectures require different decomposition strategies?

3. Training stability:
- How sensitive is the method to different initialization strategies?
- What is the impact of calibration data selection?

---

> ### Author Response · Authors · 2024-11-23
> **Response to Reviewer m9T2 (part 1)**
>
> We thank the reviewer for highlighting both the strengths and areas for improvement in our work. In the following, we answer each question with additional analysis or experiment results.
>
>
> ---
> **Q1**. *Figure 3 is confusing and needs better explanation. The caption should better explain the relationships between subfigures and clarify the flow of operations.*
>
> **A1.** We thank the reviewer for the thoughtful feedback. In the following, we provide a detailed explanation of Figure 3.
>
> * **Figure 3(a)** illustrates different components of FlatQuant, including affine transformation on weights (marked in red), per-channel scaling (marked in green), online affine transformation on activations and (de-)quantization operators (both marked in blue). Only the blue-marked components introduce inference overhead; the others are merged into LLM weights during inference.
>
> * **Figure 3(b)** demonstrates the inference dataflow and integration of FlatQuant in one LLaMA Transformer block, which is detailed in Section 3.2. For the self-attention module, $\mathbf P_a$ is applied before query, key, and value projections, while $\mathbf P_o$ is applied before the output projection. We use $\mathbf P_h$ and $\mathbf P_v$ for flattening key and value caches, respectively. Notably, $\mathbf P_v$ is merged into the value projection weights to reduce overhead. Similarly, in the feed-forward network, $\mathbf P_{ug}$ flattens the input for the up and gate projections, while $\mathbf P_d$ flattens the input for the down projection layer.
>
> * **Figure 3(c)** provides the illustration of the fast and learnable affine transformation in FlatQuant, which is introduced in Section 3.1.
>
>
> ---
> **Q2**. *While the efficient implementation and speedup is impressive, theoretical complexity analysis is crucial for algorithm development. The evaluation lacks important theoretical analysis:*
>
> * *No detailed computation complexity analysis;*
>
> * *Missing information about total parameter counts (including decomposed transform matrices);*
>
> * *No comprehensive memory consumption analysis for inference*
>
>
> **A2**. We thank the reviewer for the constructive advice. The computation complexity is analyzed in Appendix C.5. For parameter counts and memory consumption, we have included it in the revision (Line 1113~1118). Specifically,
>
> * **Computation Complexity.** As detailed in Appendix C.5 "Total FLOPs of Online Transformations" (Line 1075~1112), the total FLOPs of the online transformations in a Transformer block amount to $8bsh_d\sqrt{h_d} + 2bsh_da + 2bsh_d^2/a + 4bsh_i\sqrt{h_i}$, where $h_d$ denotes the hidden dimension, $h_i$ denotes the intermediate dimension, $a$ denotes the number of attention heads, $b$ denotes the batch size, and $s$ denotes the sequence length. For LLaMA-2-7B, the FLOPs of online transformations only account for approximately 2.61% of the total FLOPs of the FP16 model when $s=2048$.
>
> * **Total Parameter Counts and Memory Consumption.** We compute the parameter count of each online transformation below: (1) $\mathbf{P}_a: 2 (\sqrt{h_d})^2$; (2) $\mathbf{P}_o: a^2$; (3) $\mathbf{P}_h: (h_d / a)^2$; (4) $\mathbf{P}_u\ _g: 2 (\sqrt{h_d})^2$; (5) $\mathbf{P}_d: 2 (\sqrt{h_i})^2$. The total parameter count is $4h_d+2h_i+a^2+(h_d/a)^2$. The additional memory consumption during inference is $2(4h_d+2h_i+a^2+(h_d/a)^2)$ bytes, which only consumes about 0.11MB extra memory space for LLaMA-2-7B.
>
>
> ---
> **Q3**. *Citation gap for learnable clipping threshold work, such as "Duanmu H, Yuan Z, Li X, et al. SKVQ: Sliding-window Key and Value Cache Quantization for Large Language Models[J]. arXiv preprint arXiv:2405.06219, 2024."*
>
> **A3**. We thank the reviewer for the recommendation regarding related works. We have incorporated citations for SKVQ and other works on learnable clipping thresholds in Appendix A of the revised manuscript (Line 712~715).

---

> ### Author Response · Authors · 2024-11-23
> **Response to Reviewer m9T2 (part 2)**
>
> **Q4**. *Regarding the Kronecker decomposition: What is the perplexity performance without decomposition (direct training)?*
>
>
> **A4**. Compared with directly training full-size transformation matrix, Kronecker decomposition achieves similar accuracy on downstream tasks while significantly reducing inference latency.
>
> * As shown in Figure 5, different decomposition sizes have minimal impact on perplexity. However, the choice of decomposition size significantly affects inference efficiency.
>
> * To further validate the impact of decomposition on accuracy, we also conduct experiments on LLaMA-3-8B with and without decomposition. The results are summarized below.
>
> | **Method**        | **WikiText-2** | **C4** | **ARC-C** | **ARC-E** | **HellaSwag** | **LAMBADA** | **PIQA** | **Winogrande** | **Avg.** |
> | :---------------- | :------------- | :----- | :-------- | :-------- | :------------ | :---------- | :------- | :------------- | :------- |
> | w/o Decomposition | 6.88           | 11.23  | 48.98     | 76.77     | 75.92         | 73.61       | 79.54    | 71.11          | 70.99    |
> | w/ Decomposition  | 6.98           | 11.13  | 50.00     | 75.80     | 76.80         | 72.91       | 79.16    | 72.69          | 71.23    |
>
>
> ---
> **Q5**. *Model generalization:*
>
> * *How does the method perform on other model architectures (Gemini, Qwen, etc.)?*
>
> * *Are there any architectural limitations or requirements?*
>
> * *Would different model architectures require different decomposition strategies?*
>
>
> **A5**. FlatQuant is broadly applicable to different Transformer-based architectures without any architectural limitation or specific requirement.
>
> * To further validate the generality of FlatQuant, we conduct experiments on the Qwen-2.5-Instruct models. FlatQuant achieves near-lossless quantization (e.g., only 0.21% accuracy loss on QA tasks for Qwen-2.5-Instruct-32B). The results on language modeling and QA benchmarks are summarized below and included in Table 11 of the revision.
>
> | **Qwen-2.5-Instruct-7B** | **W Quantizer** | **WikiText-2** | **C4**    | **ARC-C** | **ARC-E** | **HellaSwag** | **LAMBADA** | **PIQA** | **Winogrande** | **Avg.**  |
> | :----------------------- | :-------------- | :------------- | :-------- | :-------- | :-------- | :------------ | :---------- | :------- | :------------- | :-------- |
> | FP16                     | -               | 8.36           | 14.37     | 51.37     | 75.80     | 79.57         | 67.61       | 80.20    | 69.93          | 70.75     |
> | FlatQuant                | RTN             | **8.46**       | **13.94** | 51.71     | 77.69     | 78.42         | 57.46       | 76.93    | 69.53          | **68.62** |
>
> | **Qwen-2.5-Instruct-32B** | **W Quantizer** | **WikiText-2** | **C4**    | **ARC-C** | **ARC-E** | **HellaSwag** | **LAMBADA** | **PIQA** | **Winogrande** | **Avg.**  |
> | :------------------------ | :-------------- | :------------- | :-------- | :-------- | :-------- | :------------ | :---------- | :------- | :------------- | :-------- |
> | FP16                      | -               | 5.32           | 10.45     | 58.62     | 77.02     | 85.25         | 75.14       | 81.39    | 73.16          | 75.10     |
> | QuaRot                    | RTN             | 6.95           | 12.17     | 52.13     | 74.37     | 80.41         | 68.37       | 78.45    | 67.72          | 70.24     |
> | QuaRot                    | GPTQ            | 6.54           | 11.65     | 56.06     | 76.52     | 81.83         | 71.26       | 78.78    | 69.06          | 72.25     |
> | FlatQuant                 | RTN             | **5.80**       | **10.86** | 58.62     | 78.58     | 83.72         | 75.26       | 80.74    | 72.45          | **74.89** |

---

> ### Author Response · Authors · 2024-11-23
> **Response to Reviewer m9T2 (part 3)**
>
> **Q6**. *How sensitive is the method to different initialization strategies?*
>
>
> **A6**. FlatQuant is generally robust to different initialization strategies.
>
> * By default, the initialization of online transformations is done with random orthogonal matrices. Using five different random seeds for initialization, the standard deviations of WikiText-2 perplexity (PPL) and QA accuracy are 0.008 and 0.24%, respectively, demonstrating FlatQuant's robustness to random seeds.
>
> * We further experiment with different initialization strategies, including initializing with identity matrices and random invertible matrices. The results below show that these initializations do not affect the results heavily.
>
> | **Init**                 | **WikiText-2** | **C4** | **ARC-C** | **ARC-E** | **HellaSwag** | **LAMBADA** | **PIQA** | **Winogrande** | **Avg.** |
> | :----------------------- | :------------- | :----- | :-------- | :-------- | :------------ | :---------- | :------- | :------------- | :------- |
> | Identity Matrix          | 6.98           | 11.15  | 49.49     | 77.10     | 76.82         | 74.21       | 80.09    | 70.32          | 71.34    |
> | Random Invertible Matrix | 6.97           | 11.14  | 50.60     | 77.69     | 76.35         | 72.42       | 78.89    | 70.80          | 71.12    |
> | Random Orthogonal Matrix | 6.98           | 11.13  | 50.00     | 75.80     | 76.80         | 72.91       | 79.16    | 72.69          | 71.23    |
>
>
> ---
> **Q7**. *What is the impact of calibration data selection?*
>
> **A7**. FlatQuant is robust to calibration data selection. Ablation studies presented in Appendix C.3 (Table 15) demonstrate consistent performance across various tasks with different calibration data. A summary of the results is provided below:
>
> | **Calibration Set** | **WikiText-2** | **C4** | **ARC-C** | **ARC-E** | **HellaSwag** | **LAMBADA** | **PIQA** | **Winogrande** | **Avg.** |
> | :------------------ | :------------- | :----- | :-------- | :-------- | :------------ | :---------- | :------- | :------------- | :------- |
> | WikiText-2          | 6.98           | 11.13  | 50.00     | 75.80     | 76.80         | 72.91       | 79.16    | 72.69          | 71.23    |
> | C4                  | 7.04           | 11.05  | 50.34     | 75.38     | 76.74         | 73.28       | 78.67    | 71.82          | 71.04    |
> | Pile                | 7.04           | 11.08  | 51.11     | 77.36     | 76.63         | 72.37       | 78.94    | 70.56          | 71.16    |

---

> > ### Comment · Reviewer_m9T2 · 2024-12-01
> >
> > Thank you for your response. I will maintain my score for accepting the paper.

---

### Author Response · Authors · 2024-11-23
**The General Response**

We sincerely thank all reviewers for their insightful comments and suggestions.

We would like to highlight that **FlatQuant is a lightweight yet highly effective method for LLM quantization**. It achieves near-lossless W4A4 quantization results for several models (e.g., < 1% accuracy drop for LLaMA-3-70B) with minimal latency overhead (e.g. < 0.10x speedup loss), which is valuable for deploying W4A4-quantized LLMs in real-world applications.

In response to the reviewers' feedback, we have carefully revised the manuscript (with revisions highlighted in blue). Below is a summary of the main updates:

* **More Theoretical Analysis**. We provide a more comprehensive theoretical analysis of the memory overhead of pre-quantization transformations in FlatQuant, now included in Appendix C.5 (Line 1113~1118). In short, the additional affine transformation are pretty lightweight: for LLaMa-2 7B, they only account for 2.6% FLOPs of the forward computation, and take 0.11MB in size.

* **More Experimental Results**.

  * Speedup of FlatQuant w/o kernel fusion in Figure 6. FlatQuant is on par with QuaRot in terms of latency even without kernel fusion, thanks to kronecker decomposition.

  * Experiments on Qwen models in Table 11.

  * The flatness visualizations of more linear layers in Figure 16-17.

  * A more detailed discussion of the relationship between flatness and FlatQuant in Appendix C.1 (Line 906~936).

* **Others**. We add more related works about learnable clipping in Appendix A (Line 712~715).

---

### Author Response · Authors · 2024-12-04
**The General Response (part 2)**

As the rebuttal period comes to a close, we would like to sincerely thank all the reviewers for their valuable comments and the time they dedicated to evaluating our paper. We are encouraged to see that all the reviewers recognized the contribution of our work, particularly in:

- introducing optimized decomposed affine transformations to enhance the flatness of weights and activations for effective low-bit LLM quantization [Reviewers m9T2, YdzG, tWUe];
- the speedup achieved through Kronecker decomposition and the fusion kernel [Reviewers m9T2, YdzG, RQEg, X7ak, tWUe];
- and our achievement of state-of-the-art W4A4 quantization results, demonstrated through extensive experiments [Reviewers m9T2, YdzG, X7ak, tWUe].

As Reviewer RQEg raised the concern about the latency comparison, and Reviewer X7ak raised a concern regarding the speedup without kernel fusion, we have provided thorough discussions. The results show that even without kernel fusion, the additional overhead is comparable to QuaRot/SpinQuant with superior accuracy improvements. With the fusion kernel, we achieve the best practices in W4A4 quantization for LLMs, offering both accuracy improvements and practical speedup. We believe our experiments to be fair and comprehensive.

As the deadline is approaching, we are sincerely looking forward to the following-up feedback and re-consider your evaluations.

Thanks
Authors of FlatQuant

---

### Meta-Review · Area_Chair_N8y1 · 2024-12-10

**Metareview:**

This paper explores a post-training quantization method for LLMs. The main idea is to learn a linear transformation that can be applied to the weights and activations of LLMs, aiming to reduce activation outliers and improve flatness. Additionally, the authors propose a Kronecker-product-based approach to lower computational costs; they also introduce learnable clipping thresholds.

Main strengths:
- Solid results on LLM quantization.

Main weaknesses:
- Reviewers mentioned several related papers in this area (refer to individual review comments). It appears that the proposed method might be overly incremental and lacks sufficient and fair comparisons with these existing approaches.
- Several reviewers raised concerns that the proposed approach could introduce excessive computational overhead.
- The flatness achieved through the method requires more thorough explanation. At present, it seems to rely on experimental observations without adequate theoretical justification.

**Additional Comments On Reviewer Discussion:**

Most reviewers engaged actively in the discussion. We also had an inner discussion on this paper.

See main weaknesses for the points mentioned by the reviewers.

---

### Decision · Program_Chairs · 2025-01-22

Reject